# Improved representation of plant functional types and physiology in the Joint UK Land Environment Simulator (JULES v4.2) using plant trait information

Anna Harper[1], Peter Cox[1], Pierre Friedlingstein[1], Andy Wiltshire[2], Chris Jones[2], Stephen Sitch[3], Lina M. Mercado[3,4], Margriet Groenendijk[3], Eddy Robertson[2], Jens Kattge[5], Gerhard Bönisch[5], Owen K. Atkin[6], Michael Bahn[7], Johannes Cornelissen[8], Ülo Niinemets[9,10], Vladimir Onipchenko[11], Josep Peñuelas[12,13], Lourens Poorter[14], Peter B. Reich[15,16], Nadjeda A. Soudzilovskaia[17], Peter van Bodegom[17]

**Affiliations:**
[1]College of Engineering, Mathematics, and Physical Sciences, University of Exeter, Exeter, UK
[2]Met Office Hadley Centre, Exeter, UK
[3]College of Life and Environmental Sciences, University of Exeter, Exeter, UK
[4]Centre for Ecology and Hydrology, Wallingford, UK
[5]Max Planck Institute for Biogeochemistry, Jena, Germany
[6]ARC Centre of Excellence in Plant Energy Biology, Research School of Biology, Australian National University, Canberra, Australia
[7]Institute of Ecology, University of Innsbruck, Austria
[8]Systems Ecology, Department of Ecological Science, Vrije Universiteit, Amsterdam, The Netherlands
[9]Institute of Agricultural and Environmental Sciences, Estonian University of Life Sciences, Tartu, Estonia
[10]Estonian Academy of Sciences, Tallinn, Estonia
[11]Department of Geobotany, Moscow State University, Moscow 119234, Russia;
[12]CSIC, Global Ecology Unit CREAF-CSIC-UAB, Cerdanyola del Vallès, 08193 Barcelona, Catalonia, Spain
[13]CREAF, Cerdanyola del Vallès, 08193 Barcelona, Catalonia, Spain
[14]Forest Ecology and Forest Management Group, Wageningen University, P.O. Box 6700 AA, Wageningen, The Netherlands
[15]Department of Forest Resources, University of Minnesota, St-Paul, Minnesota, USA
[16]Hawkesbury Institute for the Environment, University of Western Sydney, Penrith, New South Wales, Australia.
[17]Institute of Environmental Sciences, Leiden University, The Netherlands

*Correspondence to:* A.B. Harper (a.harper@exeter.ac.uk)

**Abstract**

Dynamic global vegetation models are used to predict the response of vegetation to climate change. They are essential for planning ecosystem management, understanding carbon cycle-climate feedbacks, and evaluating the potential impacts of climate change on global ecosystems. JULES (the Joint UK Land Environment Simulator) represents terrestrial processes in the UK Hadley Centre family of models and in the first generation UK Earth System Model. Previously, JULES represented five plant functional types (PFTs): broadleaf trees, needle-leaf trees, $C_3$ and $C_4$ grasses, and shrubs. This study addresses three developments in JULES. First, trees and shrubs were split into deciduous and evergreen PFTs to better represent the range of leaf lifespans and metabolic capacities that exists in nature. Second, we distinguished between temperate and tropical broadleaf evergreen trees. These first two changes result in a new set of nine PFTs: tropical and temperate broadleaf evergreen trees, broadleaf deciduous trees, needle-leaf evergreen and deciduous trees, $C_3$ and $C_4$ grasses, and evergreen and deciduous shrubs. Third, using data from the TRY database, we updated the relationship between leaf nitrogen and the maximum rate of carboxylation of Rubisco ($V_{cmax}$), and updated the leaf turnover and growth rates to include a trade-off between leaf lifespan and leaf mass per unit area.

Overall, the simulation of gross and net primary productivity (GPP and NPP, respectively) is improved with the 9 PFTs when compared to Fluxnet sites, a global GPP data set based on Fluxnet, and MODIS NPP. Compared to the standard 5 PFTs, the new 9 PFTs simulate a higher GPP and NPP, with the exception of $C_3$ grasses in cold environments and $C_4$ grasses which were previously over-productive. On a biome-scale, GPP is improved for all eight biomes evaluated and NPP is improved for most biomes – the exceptions being the tropical forests, savannahs, and extratropical mixed forests where simulated NPP is too high. With the

new PFTs, the global present-day GPP and NPP are 128 Pg C yr$^{-1}$ and 62 Pg C yr$^{-1}$, respectively. We conclude that the inclusion of trait-based data and the evergreen/deciduous distinction has substantially improved productivity fluxes in JULES, in particular the representation of GPP. These developments increase the realism of JULES, enabling higher confidence in simulations of vegetation dynamics and carbon storage.

**Introduction**

The net exchange of carbon dioxide between the vegetated land and the atmosphere is predominantly the result of two large and opposing fluxes: uptake by photosynthesis and efflux by respiration from soils and vegetation. $CO_2$ can also be released by land ecosystems due to vegetation mortality resulting from human and natural disturbances, such as changes in land use practices, insect outbreaks, and fires. Vegetation models are used to quantify many of these fluxes, and the evolution of the terrestrial carbon sink strongly affects future greenhouse gas concentrations in the atmosphere (Friedlingstein et al., 2006, 2015; Arora et al., 2013). A subset of vegetation models also predict both compositional and biogeochemical responses of vegetation to climate change (dynamic global vegetation models, DGVMs), one of these being the Joint UK Land Environment Simulator (JULES). JULES was the land component of the UK Hadley Centre Earth System Model (ESM) (Best et al., 2011; Clark et al., 2011), and evolved from the Met Office Surface Exchange Scheme (MOSES: Cox et al., 1998, 1999; Essery et al., 2001). Within JULES, the TRIFFID model (Top-down Representation of Foliage and Flora Including Dynamics; Cox et al., 2001) predicts changes in biomass and the fractional coverage of five plant functional types (PFTs: broadleaf trees, needle leaf trees, $C_3$ grass, $C_4$ grass, and shrubs) based on cumulative carbon fluxes and a predetermined dominance hierarchy. DGVMs such as JULES are essential for planning ecosystem management, understanding carbon cycle-climate feedbacks, and evaluating the potential impacts of climate change on global ecosystems. However, the use of DGVMs in ESMs is relatively rare. For example, in the nine coupled carbon cycle-climate models evaluated by Arora et al. (2013), only three distinct DGVMs interactively simulated changes in the spatial distribution of PFTs (the SEIB-DGVM, JSBACH, and JULES/TRIFFID).

JULES predecessor, MOSES2.2 (Essery et al., 2003), represented the land surface in the HadGEM2-ES and HadGEM2-CC simulations, and JULES will represent the land surface in the next generation UKESM. Previous benchmarking studies of JULES and MOSES identified certain areas needing improvement, such as: the seasonal cycle of evaporation, GPP, and total respiration in regions with seasonally frozen soils and in the tropics; too high growing season respiration; and too low GPP in temperate forests (Blyth et al., 2011); and too high GPP in the tropics (20°S-20°N) (Blyth et al., 2011; Anav et al., 2013). In 21st Century simulations, JULES vegetation carbon was sensitive to climate change. In particular, the tropics were very sensitive to warming, with large simulated losses of carbon stored in the Amazon forest when the climate became very dry and hot (Cox et al., 2000, 2004, 2013; Galbraith et al., 2010; Huntingford et al., 2014).

Based on these previous results, our study addressed three potential improvements in the parameterisation and representation of PFTs in JULES. First, the original five PFTs (Table 1) did not represent the range of leaf lifespans and metabolic capacities that exists in nature, and so trees and shrubs were split into deciduous and evergreen PFTs. In a broad sense, the differences between evergreen and deciduous strategies can be summarized in a leaf economics spectrum, where leaves employ trade-offs in their nitrogen use (Reich et al., 1997; Wright et al., 2004) (Fig. 1). When photosynthesis is limited by $CO_2$, the photosynthetic capacity of a leaf is dependent on the maximum rate of carboxylation of Rubisco ($V_{cmax}$). Plants allocate about 10-30% of their nitrogen into synthesis and maintenance of Rubisco (Evans 1989), while a portion of the remaining nitrogen is put toward leaf structural components; hence the strong relationship between photosynthetic capacity and leaf nitrogen concentration (e.g. Meir et al., 2002; Reich et al., 1998a; Wright et al., 2004) and leaf structure (Niinemets 1999). On average, evergreen species have a lower photosynthetic

capacity and respiration per unit leaf mass (Reich et al., 1997; Wright et al., 2004; Takashima et al., 2004), higher leaf mass per unit area (LMA) (Takashima et al., 2004; Poorter et al., 2009), allocate a lower fraction of leaf N to photosynthesis (Takashima et al., 2004), and

exhibit lower N loss at senescence (Aerts 1995; Silla and Escudero 2003; Kobe et al., 2005) than deciduous species. There is also a positive relationship between LMA and leaf lifespan (Reich et al., 1992, 1997; Wright et al., 2004). Leaves with high nutrient concentration tend to have a short lifespan and low LMA. They are able to allocate more nutrients to photosynthetic machinery to rapidly assimilate carbon at a relatively high rate (but they also

have high respiration rates). Conversely, leaves with less access to nutrients use a longer-term investment strategy, allocating nutrients to structure, defence, and tolerance mechanisms. They tend to have longer life spans, low assimilation and respiration rates, but high LMAs.

Second, we distinguished between tropical broadleaf evergreen trees and broadleaf evergreen

trees from warm-temperate and Mediterranean climates, based on fundamental differences in leaf traits, chemistry, and metabolism (Niinemets et al., 2007; Xiang et al., 2013; Niinemets et al., 2015). For example, measured $V_{cmax}$ for a given leaf N per unit area ($N_A$) can be lower in tropical evergreen trees than in temperate broadleaf evergreen trees (Kattge et al., 2011), resulting in lower $V_{cmax}$ and maximum assimilation rates for tropical forests (Carswell et al.,

2000; Meir et al., 2002, 2007; Domingues et al., 2007, 2010; Kattge et al., 2011). Collectively, the evergreen/deciduous and tropical/temperate distinctions resulted in a new set of nine PFTs for JULES: tropical broadleaf evergreen trees (BET-Tr), temperate broadleaf evergreen trees (BET-Te), broadleaf deciduous trees (BDT), needle-leaf evergreen trees (NET), needle-leaf deciduous trees (NDT), $C_3$ grasses, $C_4$ grasses, evergreen shrubs (ESh),

and deciduous shrubs (DSh) (Table 2).

Last, several parameters relating to variation in photosynthesis and respiration have not been updated since MOSES was developed in the late 1990's. We used data on LMA (kg m$^{-2}$), leaf N per unit mass, $N_m$ (kg N kg$^{-1}$), and leaf lifespan from the TRY database (Kattge et al., 145 2011; accessed Nov. 2012). The new parameters for leaf nitrogen and LMA were used to calculate a new $V_{cmax}$ at 25°C, and to update phenological parameters that determine leaf lifespan. Other parameters related to leaf dark respiration, canopy radiation, canopy nitrogen, stomatal conductance, root depth, and temperature sensitivities of $V_{cmax}$ were revised based on a review of recently available observed values, which are described in Section 2.

The purpose of our paper is to document these changes, and to evaluate their impacts on the ability of JULES to model $CO_2$ exchange for selected sites and globally on the scale of biomes, with a focus on the gross and net primary productivity. Specifically, we explore the consequences for carbon fluxes on seasonal and annual timescales of switching from the 155 current 5 PFTs to a greater number of PFTs (9) that account for growth habit (evergreen versus deciduous) and temperate/tropical plant types.

## 2. Model description

Full descriptions of the model equations are in Clark et al. (2011) and Best et al. (2011). Here 160 we briefly describe relevant current equations in JULES, associated changes in terms of updated parameter values, and document new equations and parameters. The revisions discussed in our study fall into three categories: 1) Changes to model physiology based on leaf trait data from TRY; 2) adjustment of parameters to account for the properties of the new PFTs (evergreen/deciduous, tropical/temperate); and 3) calibration of parameters based on 165 known biases in the model and a review of the literature. Parameters for the standard 5 PFTs and for the new 9 PFTs are given in Tables 1 and 2, respectively, and a summary of all

parameters are in Table SM1. For the site-level simulations, we incrementally made changes to the model to determine whether or not changes improved the simulations. This resulted in a total of eight experiments (Table 3). The version of JULES with 5 PFTs (Experiment 0) is kept as similar as possible to the configuration used in the TRENDY experiments, which are a set of historical simulations to quantify the global carbon cycle (e.g. Le Quéré et al., 2014; Sitch et al., 2015) that have been included in several recent publications. In the supplement, we provide a set of recommended parameters and guidance for users who wish to run JULES with the original 5 PFTs (Table SM2).

## 2.1 JULES model

In JULES, leaf level photosynthesis for $C_3$ and $C_4$ plants (Collatz et al., 1991, 1992) is calculated based on the limiting factor of three potential photosynthesis rates: $W_l$ (light limited rate), $W_e$ (transport of photosynthetic products for $C_3$ and PEPCarboxylase limitation for $C_4$ plants), and $W_c$ (Rubisco limited rate) (see Supplemental Material). $W_e$ and $W_c$ depend on $V_{cmax}$, the maximum rate of carboxylation of Rubisco, which is a function of the $V_{cmax}$ at 25°C ($V_{cmax,25}$):

$$V_{cmax} = \frac{V_{max,25}f_T(T_C)}{\left[1+\exp\left(0.3\left(T_C-T_{upp}\right)\right)\right]\left[1+\exp\left(0.3\left(T_{low}-T_C\right)\right)\right]} \tag{1}$$

where $T_c$ is the canopy temperature in Celcius, and

$$f_T(T_C) = Q_{10,leaf}^{0.1(T_C-25)} \tag{2}$$

$T_{upp}$ and $T_{low}$ are PFT-dependent parameters. $Q_{10,leaf}$ is 2.0.

JULES has several options for representing canopy radiation. Option 5, as described in Clark et al. (2011), includes a multi-layer canopy with sunlit and shaded leaves in each layer, two-stream radiation with sunflecks penetrating below the top layer, and light-inhibition of leaf respiration. Additionally, N is assumed to decay exponentially through the canopy with an

extinction coefficient, $k_n$, of 0.78 (Mercado et al., 2007). $V_{cmax,25}$ is calculated in each canopy layer (i) as:

$$V_{max,25,i} = n_{eff} N_{l0} e^{-k_n(i-1)/10}$$ (3)

assuming a 10-layer canopy. The parameter $N_{l0}$ is the top-leaf nitrogen content (kg N kg C$^{-1}$), and $n_{eff}$ linearly relates leaf N concentration to $V_{cmax,25}$.

Leaf dark respiration is assumed to be proportional to the $V_{cmax}$ calculated in Eq. 1:

$$R_d = f_d V_{cmax}$$ (4)

with a 30% inhibition of leaf respiration when irradiance is > 10 μmol quanta m$^{-2}$ s$^{-1}$ (Atkin et al. 2000; Mercado et al., 2007; Clark et al. 2011). Plant NPP is very sensitive to $f_d$, and since the vegetation fraction depends on NPP when the TRIFFID competition is turned on, the distribution of PFTs can also be sensitive to $f_d$. The parameter was modified from 0.015 (Clark et al., 2011) to 0.010 for all broadleaf tree PFTs in this study, based on underestimated

coverage of broadleaf trees in previous versions of JULES. Leaf photosynthesis is calculated as:

$$A_l = (W - R_d)\beta$$ (5)

where $W$ is the smoothed minimum of the three limiting rates ($W_l$, $W_e$, $W_c$), and β is a soil moisture stress factor. The factor β is 1 when soil moisture content of the root zone ($\theta$: m$^3$ m$^{-3}$) is at or above a critical threshold ($\theta_{crit}$), which depends on the soil texture. When soil water

content drops below $\theta_{crit}$, β decreases linearly until $\theta$ reaches the wilting point (where β=0) (Cox et al., 1998).

Stomatal conductance ($g_s$) is linked to leaf photosynthesis:

$$A = \frac{g_s(C_s - C_i)}{1.6}$$ (6)

where $C_s$ and $C_i$ are the leaf surface and internal $CO_2$ concentrations, respectively. The gradient in $CO_2$ between the internal and external environments is related to leaf humidity deficit at the leaf surface ($D$) following Jacobs (1994):

$$\frac{C_i - \Gamma^*}{C_s - \Gamma^*} = f_0 \left(1 - \frac{D}{D_{crit}}\right) \tag{7}$$

Here, $\boldsymbol{\Gamma^*}$ is the $CO_2$ compensation point – or the internal partial pressure of $CO_2$ at which photosynthesis and respiration balance, and $D_{crit}$ is the critical humidity deficit ($f_0$ and $D_{crit}$ are PFT-dependent parameters). In JULES, the surface latent heat flux (LE) is due to evaporation from water stored on the canopy, evaporation of water from the top layer of soil, transpiration through the stomata, and sublimation of snow. Any change to LE will also

impact the sensible heat and ground heat fluxes, since these are linked to the total surface energy balance (Best et al., 2011).

Total plant (autotrophic) respiration, $R_a$, is the sum of maintenance and growth respiration ($R_{pm}$ and $R_{pg}$, respectively):

$$R_{pm} = 0.012 R_d \left(\beta + \frac{N_r + N_s}{N_l}\right) \tag{8}$$

and

$$R_{pg} = r_g (GPP - R_{pm}) \tag{9}$$

where $r_g$ is a parameter set to 0.25 (Cox et al., 1998, 1999), and the nitrogen concentration of roots, stem, and leaves are given by $N_r$, $N_s$, and $N_l$, respectively. When using canopy radiation

model 5 in JULES, these are calculated as:

$$N_l = N_{l0}\sigma_l * LAI \tag{10}$$

$$N_r = N_{l0}\sigma_l \mu_{rl} * L_{bal} \tag{11}$$

$$N_s = N_{l0}\mu_{sl}\eta_{sl}h * L_{bal} \tag{12}$$

where $\sigma_l$ is specific leaf density (kg C m$^{-2}$ LAI$^{-1}$), $h$ is the vegetation height in meters, $L_{bal}$ is

the balanced LAI (the seasonal maximum of LAI based on allometric relationships, Cox et

al., 2001), $\mu_{rl}$ and $\mu_{sl}$ relate N in roots and stems to top-leaf N, and $\eta_{sl}$ is 0.01 kg C m$^{-1}$ LAI$^{-1}$.

In Eq. 10-12, $N_{l0}$, $\sigma_l$, $\mu_{rl}$, and $\mu_{sl}$ are PFT-dependent parameters.

The net primary productivity (NPP) is:

$$NPP = GPP - R_a \tag{13}$$

For each PFT in JULES, the NPP determines the carbon available for spreading (expanding

fractional coverage in the grid cell, only relevant when the TRIFFID competition is turned

on) or for growth (growing leaves or height). The net ecosystem exchange (NEE; positive

flux from the land to the atmosphere) is:

$$NEE = R_{eco} - GPP \tag{14}$$

where $R_{eco}$ is the total ecosystem respiration.

Phenology in JULES affects leaf growth rates and timing of leaf growth/senescence based on

temperature alone (Cox et al., 1999; Clark et al., 2011). When canopy temperature ($T_c$) is

greater than a temperature threshold ($T_{off}$), the leaf turnover rate ($\gamma_{lm}$) is equal to $\gamma_0$. When $T_c$

< $T_{off}$, the turnover rate is modified as in Eq. 15a (where $T_{off}$, $\gamma_0$, and $d_T$ are PFT-dependent

parameters):

$$\gamma_{lm} = \gamma_0\{1 + d_T(T_{off} - T_c)\} \quad for\ T_c \leq T_{off} \tag{15a}$$

$$\gamma_{lm} = \gamma_0 \qquad\qquad\qquad for\ T_c > T_{off} \tag{15b}$$

The leaf turnover rate affects phenology $\left(p = \frac{LAI}{L_{bal}}\right)$ by triggering a loss of leaf area for

$\gamma_{lm} > 2\gamma_0$, and a growth of leaf area when $\gamma_{lm} \leq 2\gamma_0$:

$$\frac{dp}{dt} = \gamma_p(1 - p) \qquad for\ \gamma_{lm} \leq 2\gamma_0 \tag{16a}$$

$$\frac{dp}{dt} = -\gamma_p \qquad\qquad for\ \gamma_{lm} > 2\gamma_0 \tag{16b}$$

where $\gamma_p$ is the leaf growth rate.


## 2.2 Updated leaf N, $V_{cmax,25}$, and leaf lifespan (Experiments 1-2)

Essentially, with the revised trait-based physiology, the parameter $\sigma_l$ (Eq. 10-11) and $N_{l0}$ (Eq. 3, 10-12) were replaced with LMA and $N_m$, respectively, from the TRY database. $N_{l0}$ and $N_m$ both describe the nitrogen content at the top of the canopy, but the former is N per unit

carbon, while the latter is the more commonly observed N per unit dry mass. $N_m$ can be converted to $N_{l0}$ using leaf carbon content per dry mass ($C_m$). Historically, $C_m$ was 0.4 in JULES (Schulze et al., 1994), but we updated it to 0.5 in all versions of JULES evaluated in this study (Reich et al., 1997; White et al., 2000; Zaehle and Friend, 2010).

We also changed the equation for $V_{cmax,25}$ from a function of $N_{l0}$ (Eq. 3) to a function of leaf N per unit area, $N_a$, a more commonly observed leaf trait, calculated as the product of the observed leaf traits LMA (kg m$^{-2}$) and $N_m$ (kg N kg$^{-1}$):

$$N_a = N_m * LMA \tag{17}$$

and $V_{cmax,25}$ ($\mu$mol $CO_2$ m$^{-2}$ s$^{-1}$) is:

$$V_{cmax,25} = i_v + s_v N_a \tag{18}$$

where parameters $i_v$ ($\mu$mol $CO_2$ m$^{-2}$ s$^{-1}$) and $s_v$ ($\mu$mol $CO_2$ gN$^{-1}$ s$^{-1}$) were taken directly from Kattge et al. (2009 – hereafter K09) (see also Medlyn et al., 1999), with two exceptions. First, the $V_{cmax}$ parameterisation from K09 was based on the leaf $C_3$ photosynthesis model. $C_4$ plants have high $CO_2$ concentration at the site of Rubisco, and therefore require less Rubisco

than $C_3$ plants (von Caemmerer and Furbank 2003). $C_4$ species typically have 30-50% as much Rubisco per unit N as $C_3$ species (Sage 1987; Makino et al., 2003; Houborg et al., 2013). We chose a slope ($s_v$) for $C_4$ to give a $V_{cmax,25}$ that is half of that for $C_3$ grass, and set

the intercept ($i_v$) to 0. This resulted in a $V_{cmax,25}$ of 32 μmol $CO_2$ m$^{-2}$ s$^{-1}$ for C4 grass, which is similar to observed values in natural grasses (Kubien and Sage 2004; Domingues et al., 2007) and $V_{cmax,25}$ in seven other ESMs (13-38 μmol $CO_2$ m$^{-2}$ s$^{-1}$; Rogers 2013). Second, K09 reported a separate $V_{cmax,25}$ for tropical trees growing on oxisols (old tropical soils with low phosphorous availability) and non-oxisols. For the BET-Tr PFT, we calculated a weighted mean slope and intercept from their Table 2 to represent an "average" tropical soil.

The new $V_{cmax,25}$ for canopy level $i$ is calculated as (replacing Eq. 3):

$$V_{max,25_i} = i_v + s_v N_a e^{-K_n(i-1)/10} \tag{19}$$

The leaf, root, and stem nitrogen contents are (replacing Eq. 10-12):

$$N_l = N_m LMA * LAI \tag{20}$$

$$N_r = N_m LMA \mu_{rl} * L_{bal} \tag{21}$$

$$N_s = \frac{N_m}{C_m} \mu_{sl} \eta_{sl} * h * L_{bal} \tag{22}$$

Four phenological parameters ($T_{off}$, $d_T$, $\gamma_0$, and $\gamma_p$, Eq. 15-16) were adjusted to capture the trade-off between leaf lifespan and LMA. We set $T_{off}$ to 5°C for deciduous trees and shrubs, to -40°C for BET-Te, NET, and ESh, and to 0°C for BET-Tr. The latter reflects the fact that many tropical evergreen tree species cannot tolerate frost (Woodward and Williams, 1987; Prentice et al., 1992). For the other evergreen PFTs, the value of -40°C ensured that plants only lose their leaves in extremely cold environments. Second, we changed $d_T$ to 0 for grasses to attain constant leaf turnover rates (Eq. 15). This fixed an unrealistic seasonal cycle in LAI of grasses and makes grasses more competitive in very cold environments (Hopcroft and Valdes, 2015). Third, we adjusted $\gamma_0$ for grasses and evergreen species to reflect the median observed leaf lifespan in the TRY database. Last, we changed $\gamma_p$ from its default value of 20 yr$^{-1}$ to 15 yr$^{-1}$ for the PFTs with the thickest leaves (NET, ESH, BET-Temp,

BET-Trop) and to 30 yr$^{-1}$ for the PFT with the thinnest leaves (DSH). The parameter $\gamma_p$

controls the rate of leaf growth in the spring and senescence at the end of the growing season

(Eq. 16b). To reduce an overestimation of uptake during the spring with the new phenology

for grass, the maximum LAI for grasses was reduced from 4 to 3.

**2.3 Other updates to JULES parameters with new PFTs (Experiments 3-6)**

Additional changes to JULES were made to account for the properties of the new PFTs, to

incorporate recent observations, and to correct known biases in the model. These fall into

four categories: radiation, stomatal conductance, photosynthesis and respiration, and plant

structure. For the site-level evaluation of JULES, we incrementally added these changes

(Table 3).

2.3.1 Stomatal conductance (Experiment 3)

JULES stomatal conductance is related to the leaf internal $CO_2$, where $C_i/C_s$ is proportional to

the parameters $f_0$ and $1/D_{crit}$ (Eq. 7). For vapour pressure deficits (D) greater than $D_{crit}$, the

stomata close. For D<$D_{crit}$, stomata gradually open in response to a reducing evaporative

demand. Needle-leaf species in JULES have a lower $D_{crit}$ than other trees, grasses, and

shrubs. The lower $D_{crit}$ increases the likelihood of the stomata being closed – similar to

Mediterranean conifers which tend to close their stomata earlier than angiosperms (Carnicer

et al., 2013) – and it tightly regulates the stomatal aperture, making plants more sensitive to

increasing D. This is analogous to plants conserving water at the expense of assimilation. We

use updated $f_0$ and $D_{crit}$ from a synthesis of water use efficiency at the Fluxnet sites (M.

Groenendijk, pers. comm.). Compared to the standard 5 PFT parameters, the $D_{crit}$ was

decreased for BET-Te, NDT, $C_3$ grass, and shrubs. The parameter $f_0$ was increased for these

PFTs, which increased $C_i$ for all D<$D_{crit}$.

2.3.2 Radiation (Experiment 4)

The light-limited photosynthesis rate ($W_l$) is proportional to α*[absorbed PAR], where α is

the quantum efficiency of photosynthesis (mol $CO_2$ [mol quanta]$^{-1}$). We reduced $\alpha$ from 0.08

to 0.06 for $C_3$ grass and evergreen PFTs typical of semi-arid and arid environments, and from

0.06 to 0.04 for $C_4$ grass, where previously the model over-predicted GPP for a given PAR.

Quantum efficiency was set at 0.10 for NDT. These values are still within the range reported

in Skillman (2008). An example of the changes is shown in the Supplemental Material, Fig.

S1. Decreasing the $\alpha$ for BET-Te and ESh PFTs helped reduce a high bias in the GPP at low

irradiances at Las Majadas (Spain – a savannah site), while increasing $\alpha$ for NDT improved

the light response of GPP at Tomakai (Japan – a Larch site).

2.3.3 Photosynthesis and respiration parameters (Experiment 5)

The leaf dark respiration is calculated as a fraction, $f_d$, of $V_{cmax}$ (Eq. 4). In testing JULES, we

found that $C_3$ grasses were overly productive and tended to be the dominant grass type even

in tropical ecosystems where we expected $C_4$ dominance. Therefore, we increased the $f_d$ for

$C_3$ (from 0.015 to 0.019) and decreased the $f_d$ for $C_4$ (from 0.025 to 0.019) so the two grass

PFTs would have similar $R_d$ rates for a given $V_{cmax}$.

Preliminary evaluation of JULES GPP at the Fluxnet sites in Table 4 revealed the need for a

higher (lower) $V_{cmax,25}$ for the BET-Tr and NDT (BET-Te) PFTs than the mean value

reported in K09. For these PFTs, the slope parameter ($s_v$) was adjusted to result in the final

$V_{cmax,25}$ for each PFT (black bars, Fig. 2), using the mean ± 1 standard deviation of $V_{cmax,25}$

from K09 as an upper limit.

$T_{upp}$ and $T_{low}$ were also modified, as optimal $V_{cmax}$ can occur at temperatures near 40°C (Medlyn et al., 2002), and the previous optimal temperature for $V_{cmax}$ was 32°C for BT and

22°C for NT. A study of seven broadleaf deciduous tree species found $T_{opt}$ for $V_{cmax}$ ranging from 35.9°C to >45°C (Dreyer et al., 2001), and maximum $V_{cmax}$ can occur at temperatures of at least 38°C in the Amazon forest (B. Kruijt, pers. comm.). Therefore, we changed $T_{opt}$ from 32°C to 39°C for all broadleaf trees and from 22°C to 33° and 32°C for NET and NDT, respectively. $C_3$ grass $T_{opt}$ was decreased from 32°C to 28°C to help reduce the high

productivity bias in grasses.

Additionally, the ratio of nitrogen in roots to leaves ($\mu_{rl}$) was updated following the relationships in Table 1 of Kerkhoff et al. (2006). However, instead of assigning a separate $\mu_{rl}$ for each PFT, we assigned the mean values for trees/shrubs and grasses (0.67 and 0.72,

respectively).

2.3.4 Plant Structure (Experiment 6)

There is evidence that larch trees (NDT) can be tall with a relatively low LAI compared to needle-leaf evergreen trees (Ohta et al., 2001; Hirano et al., 2003) and compared to broadleaf

deciduous trees (Gower and Richards 1990). In JULES, canopy height ($h$) is proportional to the balanced LAI, $L_b$:

$$h = \frac{a_{wl}}{a_{ws}*\eta_{sl}} L_b^{b_{wl}-1} \tag{23}$$

The parameter $a_{wl}$ relates the LAI to total stem biomass, and for trees it is 0.65. Hirano et al. (2003) found h=15 m and maximum LAI=2.1, which would imply $a_{wl}$ =0.91, and Ohta et al.

(2001) found h=18 m and LAI=3.7, implying $a_{wl}$=0.75. Therefore we adjusted $a_{wl}$ for NDT to 0.75, which was an important change for allowing NDT to out-compete BDT in high latitudes.

We also changed the root depths, although these changes were constrained by the 3 m deep

soil in the standard JULES setup. Previously, root depths were 3 m for broadleaf trees, 1 m

for needle-leaf trees, and 0.5 m for grasses and shrubs (Best et al. 2011). With the new PFTs,

roots are shallower for BET-Te and BDT (2 m), and deeper for NET (1.8 m), NDT (2 m), and

shrubs (1 m) (Zeng 2001).

**3. Methods**

**3.1 Data**

We analysed leaf $N_m$, specific leaf area (=1/LMA), and leaf lifespan from the TRY database

(accessed in Nov. 2012). Data was translated from species level to both the standard five and

new nine PFTs based on a look-up table provided by TRY, and screened for duplicate entries.

We only selected entries with measurements for both LMA and $N_m$. This resulted in 9,372

LMA /$N_m$ pairs and 1,176 leaf lifespan measurements (Supplemental Material).

To evaluate the model performance we used GPP from the Model Tree Ensemble (MTE) of

(Jung et al., 2011), MODIS NPP from the MOD17 algorithm (Zhao et al. 2005; Zhao and

Running, 2010), and GPP and NEE from 13 and 14 Fluxnet sites, respectively (Table 4).

Using the net exchange of $CO_2$ observed at the Fluxnet sites, NEE was partitioned into GPP

and $R_{eco}$. Assuming that night-time NEE=$R_{eco}$, $R_{eco}$ was estimated as a temperature function

of night-time NEE (Reichstein et al., 2005; Groenendijk et al., 2011).

**3.2 Model simulations**

We performed two sets of simulations to evaluate the impacts of the new PFTs in JULES

v4.2. First, site-level simulations used observed meteorology from 14 Fluxnet towers – these

include the nine original sites benchmarked in the study of Blyth et al. (2011), plus an additional five to represent more diversity in land cover types and climate. The vegetation cover was prescribed as in Table 4, and vegetation competition was turned off. The changes described in Section 2.2 and 2.3 were incrementally added to evaluate the effect of each group of changes (Table 3). Full results are shown in the Supplemental Material, but for the main text we focus the discussion on JULES with 5 PFTs (JULES5); JULES with 9 PFTs and updated $N_m$, LMA, $V_{cmax,25}$, and leaf lifespan from the TRY database (JULES9$_{TRY}$); and JULES with 9 PFTs and all updated parameters described in Section 2.3 (JULES9$_{ALL}$). These are, respectively, Experiments 0, 2, and 7 in Table 3.

Soil carbon takes more than 1000 years to equilibrate in JULES, so we used an accelerated method that only requires 200-300 years of spin up (depending on the site). JULES has four soil pools (decomposable and resistant plant material, long-lived humus, and microbial biomass), and the decomposable material pool has the fastest turnover rate (equivalent to ~10 yr$^{-1}$) (Clark et al. 2011). For each experiment, soil carbon was spun up using accelerated turnover rates in the three slower soil pools for the first 100 years. The rates of the resistant, humus, and biomass bools were increased by a factor of 33, 15, and 500, respectively, so all pools had the same turnover time as the fastest pool. This resulted in unrealistically depleted soil carbon pools. The second step of the spin up was to multiply the pool sizes by these same factors, and then allow the soil carbon to spin up under normal conditions for an additional 100-200 years.

Second, global simulations were conducted for JULES5 and JULES9$_{ALL}$. It could be argued that similar model improvements might be gained with the original five PFTs with improved parameters. We tested this hypothesis with a third global experiment, JULES5$_{ALL}$, with 5

PFTs but improved parameters (Table SM2). The global simulations followed the protocol

for the S2 experiments in TRENDY (Sitch et al., 2015), where the model was forced with

observed annual-average $CO_2$ (Dlugokencky and Tans, 2013), climate from the CRU-NCEP

data set (v4, N. Viovy, pers. comm.), and time-invariant fraction of agriculture in each grid

cell (Hurtt et al., 2011). Vegetation cover was prescribed based on the European Space

Agency's Land Cover Climate Change Initiative (ESA LC_CCI) global vegetation

distribution (Poulter et al., 2015, processed to the JULES 5 and 9 PFTs by A. Hartley) (Fig.

3a). JULES did not predict vegetation coverage in this study, which enabled us to evaluate

JULES GPP and NPP given a realistic land cover. The evaluation of vegetation cover and

updated competition for 9 PFTs will be evaluated in a follow-up paper. Since the land cover

was prescribed based on a 2010 map, we also set the agricultural mask based on land use in

2010, and enforced consistency between the two maps such that fraction of agriculture could

not exceed the fraction of grass in each grid cell. During the spin-up (300 years with 100

years of accelerated turnover rates as at the sites), we used atmospheric $CO_2$ concentration

from 1860 and recycled climate from 1901-1920. The transient simulation (with time-varying

$CO_2$ and climate) was from 1901-2012. The model spatial resolution was N96 (1.875°

longitude x 1.25° latitude).


**3.3 Model Evaluation**

The model evaluation is presented in two stages. First, using the site-level simulations, we

evaluated GPP and NEE with the root mean square error (RMSE) and correlation coefficient,

*r*, based on daily and monthly averaged fluxes, respectively. Site history can result in non-

zero annual NEE, but JULES maintains annual carbon balance, so it is not realistic to expect

the simulated annual NEE to match the observations. Therefore, we compared anomalies of

NEE instead.

We summarized the changes in RMSE and $r$ using relative improvements for each
experiment in Table 4, $i$. The statistics were calculated such that positive values denote an
improvement compared to JULES5 (Experiment 0):

$$RMSE\_rel_i = \frac{RMSE_{5pfts} - RMSE_i}{RMSE_{5pfts}} \tag{23}$$

$$r\_rel_i = \frac{r_i - r_{5pfts}}{r_{5pfts}} \tag{24}$$

Second, we compared the model from global simulations to biome-averaged fluxes in eight
biomes based on 14 World Wildlife Fund terrestrial ecoregions (Olson et al., 2001) (Fig. 3b,
Table S3). Fluxes were averaged for the land in each biome in both the model and the
observations. We evaluated seasonal cycles of GPP from the MTE (Jung et al., 2011), and
annually averaged GPP (from the MTE) and NPP (from MODIS). The tropical forest biome
includes regions of tropical grasslands and pasture – in the ESA LC_CCI data set, the BET-
Tr PFT is dominant in only 38% of the biome and grasses occupy 36%. Therefore, we only
included the grid cells where the dominant PFT in the ESA data is BET-Tr. The extratropical
mixed forest biome has a large coverage of agricultural land, and as a result 46% of the
biome is $C_3$ grass, while BDT and NET only cover 14% and 8% of the biome, respectively.
We omitted grid cells with >20% agriculture in 2012 to calculate the biome average fluxes.

## 4 Results

### 4.1 Data analysis of leaf traits

With the previous 5 PFTs, only the needle-leaf tree PFT occupied the "slow investment" end
of the leaf economics spectrum (high LMA and low $N_m$) (Fig. 1). The new PFTs were given
the median $N_m$ and LMA from the TRY dataset (Fig. 1c), and these exhibit a range of
deciduous and evergreen strategies, although there is substantial overlap between PFTs. The

needle-leaf evergreen trees, evergreen shrubs, and temperate broadleaf evergreen trees have

low $N_m$ and thick leaves, but their $N_A$ (shown in the legend of Fig. 1a,c) is relatively high (>2

g m$^{-2}$), which has been long known for species with long leaf lifespans (>1 year) (Reich et al.,

1992). These traits on aggregate indicate that they use the "slow investment" strategy of

growing thick leaves with low rates of photosynthesis per unit investment of biomass.

Compared to the evergreen PFTs, the deciduous shrubs and broadleaf deciduous trees have

higher $N_m$, thinner leaves, lower $N_A$ (1.3-1.7 g N m$^{-2}$), and leaf lifespans of less than six

months. The tropical broadleaf evergreen trees have a moderate $N_m$ and leaf thickness, with

an average lifespan of 11 months, reflecting a mixture of successional stages in the database.

The grasses have the shortest leaf lifespans. C$_4$ grasses have high LMA, low $N_m$, and a high

$N_A$; while the thinner C$_3$ grasses have a high $N_m$ and low $N_A$. Figure 1 also shows the impacts

of changing the phenological parameters ($T_{off}$, $d_T$, $\gamma_0$, and $\gamma_p$, Eq. 15-16) on median leaf

lifespan during a 30-year global simulation, where now JULES captures the observed leaf

lifespans.

Based on the new $N_A$, $V_{cmax,25}$ was updated using the new parameters $i_v$ and $s_v$ (Eq. 18; Fig.

2). The values calculated from the TRY data are shown with asterisks, and these were used in

the JULES9$_{TRY}$ experiments. The black bars show the final $V_{cmax,25}$ after adjusting $s_v$ for the

two broadleaf evergreen tree PFTs and the needle-leaf deciduous trees (see Section 2.3.3).

Within the trees, the temperate broadleaf evergreen PFT has the highest $V_{cmax,25}$, while the

needle-leaf deciduous and tropical broadleaf evergreen PFTs have the lowest. Because the

JULES C$_3$ and C$_4$ PFTs are assumed representative of natural vegetation, they have relatively

low $V_{cmax,25}$ (compared to the range from K09 for C$_3$). The $N_A$ calculated from median $N_m$ and

LMA in this study (1.19 g N m$^{-2}$) is lower than the average $N_A$ reported in K09 (1.75 g N m$^{-}$

[2]). However, the $C_3$ $V_{cmax,25}$ (51.09 μmol $CO_2$ m$^{-2}$ s$^{-1}$) is close to values reported for European grasslands (41.9±6.9 μmol $CO_2$ m$^{-2}$ s$^{-1}$ and 48.6±3.5 μmol $CO_2$ m$^{-2}$ s$^{-1}$ for graminoids and

forbs, respectively, in Wohlfahrt et al., 1999). In comparison to JULES5, the new $V_{cmax,25}$ is higher for all PFTs except for $C_3$ grass. Previously, the $V_{cmax,25}$ was lower than the observed range for all non-tropical trees, but now the $V_{cmax,25}$ for all PFTs is within the range of observed values.

**4.2 Site level simulations**

In most cases, the higher $V_{cmax}$ from trait data increased the GPP and NPP, and resulted in higher respiration fluxes due to both autotrophic (responding to higher GPP) and heterotrophic (responding to higher litterfall due to higher NPP) respiration. First, we compared JULES with 5 PFTs (JULES5) to JULES with 9 PFTs and the TRY data

(JULES9$_{TRY}$) (Experiments 1 and 2, respectively, in Table 3) at the sites listed in Table 4. The results are summarized in Figure 4, where yellows and reds indicate increased correlation (Fig. 4a, b) or reduced RMSE (Fig. 4c, d) in each experiment compared to JULES5. Using the $N_m$, LMA, and $V_{cmax,25}$ data from TRY improved the seasonal cycle of GPP at the two tropical forest sites, the evergreen savannah, and the crop site, and decreased

the daily RMSE at one NET site (Tharandt), all grass sites, and the NDT site (Tomakai) (Experiment 1, Fig. 4). Enforcing the LMA-leaf lifespan relationship further improved the seasonal cycle at both savannah sites, the two natural $C_3$ grass sites (the seasonal cycle was worse at the crop site), and the NDT site, and further reduced RMSE at the deciduous savannah site and one BDT site (Harvard) (Experiment 2, aka JULES9$_{TRY}$). In comparison,

applying all parameter changes summarized in Table 3 further reduced the RMSE at every site except the two tropical forests and further increased $r$ at every site except the tropical forests and the evergreen savannah (Experiment 7, aka JULES9$_{ALL}$).

Overall, the carbon and energy exchanges were best captured with JULES9$_{\text{ALL}}$. Compared to JULES5, the RMSE for GPP in JULES9$_{\text{ALL}}$ decreased by more than 40% at Kaamanen (C3 grass), Tharandt (NET), and Tomakai (NDT); the daily RMSE of NEE decreased at eight sites; and $r$ increased for NEE at 11 sites. The only sites without an improvement in either metric for NEE were Manaus (BET-Tr) and Bondville (Crop). The improvements to NEE were large at Tharandt (r from 0.61 to 0.76), Fort Peck C$_3$ grass (0.05 to 0.38), and Tomakai (0.09 to 0.93), and RMSE for NEE decreased by more than 35% at Kaamanen and Tomakai. Respiration and latent heat fluxes are discussed in the Supplemental Material.

On an annual basis, GPP was higher in JULES9$_{\text{ALL}}$ than in JULES5 at every site except for the Tapajós K77 pasture, El Saler (NET), Tonzi (savannah), and Kaamanen, and NPP was higher at every site except for Tapajós K77, El Saler, and Kaamanen (Table 5). Total GPP was improved at every site except for Hyytiälä (NET) and Las Majadas (savannah), where annual GPP was too high in JULES5, and at El Saler and Tonzi, where the modelled GPP was too low. However, for every site except Hyytiälä, JULES9$_{\text{ALL}}$ was within the range of observed annual GPP. We now explore some site-specific aspects of the carbon cycle results.

4.2.1 Broadleaf forests

Both GPP and NPP were higher in JULES9$_{\text{ALL}}$ than JULES5 for broadleaf forests due to a higher $T_{\text{opt}}$ of $V_{\text{cmax}}$ and a higher $V_{\text{cmax,25}}$. Simulated GPP was similar to observations in the absence of soil moisture stress. The increase in GPP occurred year-round at Manaus, but only during the wet season at Tapajós K67 (Fig. 5). GPP was similar in all JULES simulations during the dry season (Oct.-Dec.), when soil moisture deficits limited photosynthesis. The soil moisture stress factor, β, was <0.7 during these months, while it was >0.87 all year at

Manaus (recall that a higher β indicates less stress). The reduction in GPP during the dry season at both sites is in contrast to the observations, which show an increase from Aug.-Dec.

As a result, the simulated seasonal cycle of GPP was incorrect at both sites, and although the annual total GPP was closer to observations, the monthly RMSE was higher in JULES9$_{ALL}$ compared to JULES5. The simulated NPP was too low in JULES5 at both sites. In JULES9$_{ALL}$, the NPP was too high at Manaus (by 187 g C m$^{-2}$ yr$^{-1}$) and too low at Tapajós (by 396 g C m$^{-2}$ yr$^{-1}$).


At the two BDT sites (Harvard and Morgan Monroe), the peak summer GPP was closer to observations in JULES9$_{ALL}$. GPP was very well reproduced at Harvard (BDT), where the average JJA temperature was 4°C cooler than at Morgan Monroe (29°C compared to 33°C), and, due to differences in the soil parameters, the soil moisture stress factor was higher (β

>0.8 at Harvard compared to 0.5<β<0.7 at Morgan Monroe). At Morgan Monroe, the observed GPP was nearly zero from Nov.-Mar., but all versions of JULES simulated uptake during Nov.-Dec., when the average temperatures were still above freezing, possibly due to leaves staying on the trees for too long in the model. The RMSE of NEE decreased (Fig 5b), but the amplitude of the seasonal cycle was too small at both BDT sites.


4.2.2 Needle-leaf forests

The seasonal cycle of GPP improved at the needle-leaf forests, but JULES9$_{ALL}$ underestimated GPP during mid-summer at the larch site (Tomakai) and during the summer at a Mediterranean site (El Saler), and overestimated summertime GPP at a cold conifer site

(Hyytiälä). Although there was a large improvement in the seasonal cycle at El Saler in JULES9$_{ALL}$, the GPP was still underestimated during the dry months of June-Oct. During this period, β reduced to a minimum of 0.17 in August, and the GPP was too low by an average

1.83 g C m$^{-2}$ d$^{-1}$. At all sites there was shift toward stronger net carbon uptake during the summer months with the new PFTs, which increased the correlation with observed NEE. At El Saler, the RMSE of NEE increased due to a change in the seasonal cycle of leaf dark respiration (R$_d$, Eq. 8) resulting from the higher $T_{opt}$. At Hyytiälä, the RMSE of NEE increased due to higher rates of soil respiration during the winter months (Fig. S3; where soil respiration is the difference between total and autotrophic respiration).

Compared to JULES5 (with a needle-leaf PFT), both GPP and respiration were improved with the new NDT PFT at Tomakai, primarily due to an improved seasonal cycle of GPP with the deciduous phenology (Experiment 2). In JULES5, the LAI at the site was 6.0 m$^2$ m$^{-2}$, compared to a summer maximum of ~3.5 m$^2$ m$^{-2}$ with the deciduous phenology and to a reported average LAI of larch of 3.8 m$^2$ m$^{-2}$ (Gower and Richards, 1990). The new deciduous PFT also improved the seasonal cycle of NEE, and reduced errors in LE and SH (Fig. S4). The magnitude of maximum summertime GPP was still underestimated, but this could be because the site is a plantation, where trees are evenly planted to optimize the incoming radiation, rather than a natural larch forest.

### 4.2.3 Grasses

GPP and NEE were improved for temperate grasslands (Kaamanen and Fort Peck) and NEE was improved at a tropical pasture (Tapajós K77). Compared to JULES5, productivity in JULES9$_{ALL}$ was higher at a temperate C$_3$ site (Fort Peck), and lower at a cold C$_3$ site (Kaamanen) and the tropical C$_4$ site. In terms of GPP, these changes brought JULES9$_{ALL}$ closer to the observations (Table 5). With the new PFT parameters, grasses had higher year-round LAI due to the removal of phenology, and GPP increased earlier in the year at Kaamanen, Bondville, and Fort Peck in JULES9$_{ALL}$ compared to JULES5. Net uptake also

occurred 1-2 months earlier in JULES9$_{ALL}$ (compared to JULES5), which decreased RMSE and increased $r$ for NEE at the three natural grassland sites. JULES9$_{ALL}$ underestimated productivity at Bondville (crop site), but this is not surprising given that the PFT is meant to represent natural grasses. There is a separate crop model available for JULES (Osborne et al., 2015).

The Tapajós K77 pasture was not included in the set of sites with GPP/R$_{eco}$ partitioning. The simulated GPP was lower in JULES9$_{ALL}$ than in JULES5 due to the lower quantum efficiency (Fig. S3c). The seasonal cycle of NEE was close to observed during most months (Fig 5b), and in terms of $r$ and RMSE JULES9$_{ALL}$ was better than JULES5. In JULES5, the GPP and NPP were higher at the Tapajós K77 pasture than at the Tapajós K67 forest site despite being driven by the same meteorology (Table 5). In JULES9$_{ALL}$, GPP was higher at the forest site than at the pasture, and the NPP was similar.

4.2.4 Mixed vegetation sites

Las Majadas and Tonzi are savannah sites dominated by evergreen and deciduous plants, respectively (assumed in the simulations to be an equal mix of trees, shrubs, and C$_3$ grass, Table 4). Both GPP and NPP were better simulated with JULES9$_{ALL}$ at both sites, and the annual GPP was within the range of the observations (although it was too high at Las Majadas and too low at Tonzi).

At Las Majadas, the GPP increased in JULES9$_{ALL}$ (compared to JULES5) during the wet spring (Jan.-Apr.) due to high GPP from the BET-Te and C$_3$ grass PFTs. The former had a higher year-round LAI (~4.6 m$^2$ m$^{-2}$), $V_{cmax,25}$, and $T_{opt}$ for $V_{cmax}$ compared to the BT from the 5 PFTs (which had maximum summer LAI of 3.8 m$^2$ m$^{-2}$). For C$_3$ grass, the new $V_{cmax,25}$ and

$T_{opt}$ were lower in JULES9$_{ALL}$, but the removal of phenology (setting $d_T$ to 0) increased the LAI during the cool, mild winter months when photosynthesis could still occur. Grid-cell

mean GPP was also slightly higher during the hot, dry summer, again owing to the BET-Te PFT. The simulated seasonality NEE was similar to observations ($r$=0.70), but the April-May uptake was too strong and resulted in an overestimation of the annual GPP.

At Tonzi, GPP was similar to observations except during April-July, when it was too low.

The modelled photosynthesis began to decline after March, coinciding with a rapid increase in simulated soil moisture stress and stomatal resistance. Moving from a generic to a deciduous shrub resulted in a large decrease in simulated GPP at this site. The shrub LAI decreased from ~3.3 m$^2$ m$^{-2}$ to a maximum of 1.5 m$^2$ m$^{-2}$, and the $V_{cmax,25}$ for the DSh was slightly lower than the $V_{cmax,25}$ for the generic shrub. Slightly compensating for the lower

shrub GPP was a higher broadleaf tree GPP, with a higher $V_{cmax,25}$ and $T_{opt}$ compared to the previous values in JULES5.

**4.3 Global results**

In this section, we analyse the impact of the PFT-specific biases and improvements on

biome-scale GPP and NPP fluxes in global simulations. The area-weighted fluxes are displayed in Table 6 and Figure 6 for the biomes shown in Fig. 3, and seasonal cycles are shown in Figure 7. GPP increased in JULES9$_{ALL}$ compared to JULES5 in all extratropical biomes, but it decreased in the two biomes with significant coverage by C$_4$ grass. For all biomes, the representation of GPP in JULES9$_{ALL}$ was closer to the observed (MTE) value.

NPP increased in every biome, and this was an improvement (relative to MOD17) in five biomes (boreal and coniferous forests, temperate grasslands, deserts/shrublands, tundra, and

Mediterranean woodlands), but NPP was too high in tropical biomes and extratropical mixed forests.

In the tropical forests, the biome-average GPP and NPP increased in JULES9$_{\text{ALL}}$ compared to JULES5, and both fluxes were ~200 g C m$^{-2}$ yr$^{-1}$ higher than their respective observational value. The seasonality of rainfall in the Tropics has a hemispheric dependence. Splitting the biome into the northern and southern hemisphere revealed that the seasonal cycle in Fig. 7a was most similar to the southern hemisphere in terms of the climate and fluxes. In both

hemispheres, the JULES GPP was higher than the MTE GPP during the transition period from the wet to the dry season and the early dry season. This is in contrast to the results at the two Brazilian Fluxnet sites, where JULES GPP was lower than observed during the dry ~~wet~~ season.

Most of the differences between JULES5$_{\text{ALL}}$ and JULES9$_{\text{ALL}}$ were in the tropics (Fig. 9, Table 6). The global GPP was relatively high (135 Pg C yr$^{-1}$) in JULES5$_{\text{ALL}}$ (compared to 127 Pg C yr$^{-1}$ for JULES9$_{\text{ALL}}$), primarily because $V_{\text{cmax}}$ for the generic broadleaf tree was much higher than for the tropical broadleaf evergreen PFT, based on the data from K09. Although tropical GPP was higher in JULES5$_{\text{ALL}}$ compared to JULES9$_{\text{ALL}}$, the NPP in

tropical forests was lower and closer to the values from MODIS NPP. The reason was the differences in leaf nitrogen, which increased respiratory costs in JULES5$_{\text{ALL}}$ compared to JULES9$_{\text{ALL}}$. Both N$_{\text{A}}$ and N$_{\text{m}}$ were higher for the broadleaf tree PFT than for the tropical evergreen broadleaf tree PFT.

Over the tropical savannah biome, the GPP decreased in JULES9$_{\text{ALL}}$ compared to JULES5 due to lower productivity from C$_4$ grasses, and GPP was within the uncertainty range of the

MTE GPP, although slightly higher. The overestimation occurred during most of the year (Fig. 7b), except during the late dry season/early wet season (Oct.-Dec.). Although $C_4$ grasses had a lower NPP in JULES9$_{ALL}$, a significant fraction of the biome is composed of $C_3$ grass, BDT, ESh, and DSh in the ESA data, which all had higher NPP in JULES9$_{ALL}$. For this reason, biome-scale NPP was higher in JULES9$_{ALL}$ than in JULES5, and simulated NPP was 140 g C m$^{-2}$ yr$^{-1}$ higher than the MOD17 value. In the temperate grasslands biome, both GPP and NPP were higher in JULES9$_{ALL}$ compared to JULES5, and closer to the MTE and MOD17 values. However, compared to the MTE, the JULES9 GPP increased one month early, it was too low in the mid-summer, and it declined too slowly in the autumn.

The biome-scale GPP in the extratropical mixed forests improved in JULES9$_{ALL}$ compared to JULES5, and was very close to the MTE estimate. The simulated GPP was overestimated during the autumn (Sept.-Oct.) and underestimated during the winter. Simulated NPP was very close to the MOD17 NPP in JULES5, but it is too high by ~100 g C m$^{-2}$ yr$^{-1}$ in JULES9$_{ALL}$. The predominant vegetation types in the "boreal and coniferous forests" biome are NET (26% coverage), $C_3$ grass (20%), and NDT (14%). Shrubs, deciduous broadleaf trees, and bare soil cover the remaining 40% of the biome. There was a large increase in summertime GPP in this biome, bringing JULES9$_{ALL}$ closer to the MTE GPP than JULES5. The NPP increased in JULES9, compared to JULES5, and was within 10 g C m$^{-2}$ yr$^{-1}$ of the MOD17 NPP.

Deserts/shrublands and tundra are both dry environments with annual average GPP of ~280 g C m$^{-2}$ yr$^{-1}$ according to the MTE dataset. Although GPP increased in both biomes in JULES9$_{ALL}$ relative to JULES5, it was much lower than the MTE value. In the tundra biome, GPP was underestimated during the entire growing season, and it was underestimated all year

in the desert biome. The simulated NPP was also significantly lower than MOD17 in these two biomes, although it was slightly improved in JULES9$_{ALL}$. These results indicate that the JULES plants struggle in extremely cold and arid environments.


In the Mediterranean woodlands, GPP increased by 90 g C m$^{-2}$ yr$^{-1}$ and NPP increased by 80 g C m$^{-2}$ yr$^{-1}$ in JULES9$_{ALL}$ compared to JULES5, but both fluxes were still ~100 g C m$^{-2}$ yr$^{-1}$ lower than the MTE GPP and MOD17 NPP. The simulated GPP (in JULES9$_{ALL}$) was close to the MTE value during most of the year except the dry season, when it declined more in the

model than in the MTE estimate.

On a global scale, JULES9$_{ALL}$ had a similar GPP but higher NPP compared to JULES5 (Fig. 8). In both simulations, the global GPP was 128-129 Pg C yr$^{-1}$ (average from 2000-2012), compared to the MTE average of 122±8 Pg C yr$^{-1}$. GPP was higher in JULES9$_{ALL}$ compared

to JULES5 in the core of the tropical forests, but lower in tropical/subtropical South America, Africa, and Asia. These are regions with significant grass coverage (Fig. 3a), especially C$_4$ grasses. Poleward of 30°, GPP was higher in JULES9$_{ALL}$ due to higher productivity in trees. In JULES5, the global NPP (55 Pg C yr$^{-1}$) was close to the value from MODIS NPP (54 Pg C yr$^{-1}$). In JULES9$_{ALL}$, the NPP was higher than JULES5 almost everywhere (except for

southern Brazil where C$_4$ grasses are dominant), and the global NPP was 62 Pg C yr$^{-1}$.

**5. Discussion**

**5.1 Impacts of trait-based parameters and new PFTs**

Including trait-based data on leaf N, $V_{cmax,25}$, and leaf lifespan improved the seasonal cycle of

GPP at seven sites, especially sites with C$_3$ grass and NDT. Parameterizing leaf lifespan correctly has been shown to be important, even within biomes (Reich et al., 2014). Our study

confirms this, as the simulation of GPP improved at fewer sites in the simulations without the improved leaf lifespan. However, compared to the standard 5 PFTs, the RMSE of GPP was only improved at four sites in JULES9$_{TRY}$. Despite this, the new PFTs with the new trait data

include observed trade-offs between leaf structure and lifespan. These trade-offs are important for enabling JULES to represent observed vegetation distribution and for predictions of future fluxes.

Incorporating more data and accounting for evergreen and deciduous habits further improved

the model, as indicated by the closer model-data comparison obtained with JULES9$_{ALL}$ at both the site and global level. The distinction between the tropical and temperate broadleaf evergreen trees provided mixed results. While there was no improvement in the seasonal cycles at the two tropical forest sites, both GPP and the seasonal cycle of NEE were improved at the warm-temperate evergreen savannah site (Las Majadas). This study has laid

the groundwork for further improvements to JULES GPP and plant respiration by incorporating trait-based physiological relationships and allowing for a flexible number of PFTs. Future development can focus on more biome-specific data-model mismatches than was possible with the generic set of 5 PFTs.

The 9 PFTs were chosen as they represent the range of deciduous and evergreen plant types with minimal externally determined bioclimatic limits. The distinction between tropical and temperate broadleaf evergreen trees account for the important differences between these types of trees (e.g. a lower $V_{cmax}$ for a given $N_A$ in tropical broadleaf evergreen trees: Kattge et al., 2009). The comparison of JULES5$_{ALL}$ and JULES9$_{ALL}$ indicates that even using

improved parameters with 5 PFTs based on the TRY data and the literature reviewed in this study will give improved productivity fluxes in JULES. However, an important caveat is

JULES was not run with dynamic vegetation for this analysis. The additional PFTs enable more diverse and specific dynamic responses to climate change.

**5.2 Future development priorities**

The biome-level evaluation of GPP and NPP provides insight into potential areas for improvement in JULES: in particular boreal forests, tundra, Mediterranean woodlands and desert/xeric shrublands (Fig. SM6). GPP was systematically underestimated in regions experiencing seasonal soil moisture stress, such as the tropical forests, summer at Morgan

Monroe, and the dry season at El Saler. A similar result was seen with the arid biomes and in the Mediterranean biome during summer. The fact that the model did not match the seasonal cycle of GPP at the two tropical forest sites with improved parameters indicates that processes such as the representation of plant water access and/or soil hydraulic properties need to be addressed in JULES. However, the dry season bias was not present when JULES

was compared to the biome-scale MTE GPP. This underscores the complexity of modelling tropical forest productivity and the need to evaluate multiple data sources. High latitude grasses were underproductive, which also contributed to an underestimation of soil carbon (not shown). Further development of a tundra-specific PFT(s) could improve the carbon cycle in these regions.


A side effect of the trait-based parameters was increased respiration, and comparison to both Fluxnet sites and the MTE suggest it is now too high for most biomes. Total ecosystem respiration was higher than observed at Manaus, Harvard, Morgan Monroe, Tharandt, Hyytiälä, Kaamanen, Las Majadas, and Tonzi (75% of the sites with respiration data) (Fig.

S3). As this study has focused primarily on improving the GPP, the next step should be to include a more mechanistic representation of growth and maintenance respiration in JULES

to improve the net productivity (e.g. using data from Atkin et al., 2015). Comparison to the MTE respiration also suggests that JULES soil respiration is too high during the winter in the temperate and boreal biomes. In the latter, both versions of JULES predicted positive

respiration flux during the winter, while the MTE product showed negligible fluxes (Fig. S5). The average winter temperatures in the biome were <-13°C, yet soil respiration continued during these months because the Q10 soil respiration scheme has a very slow decay of soil respiration flux at sub-zero temperatures (see Fig. 2 of Clark et al., 2011). A similar result was seen at Hyytiälä (Fig. S3b), which further indicates that winter-time respiration might be

too high.

Last, the simulation of GPP could be further improved by replacing the static $V_{cmax,25}$ per PFT. Simultaneous with this study, there is work to include temperature acclimation for photosynthesis JULES, which is more realistic than a set $T_{opt}$ for each PFT. Also, the data

exhibits large within-PFT variation in $V_{cmax,25}$ (Fig. 2) and photosynthetic capacity can depend on the time of year. Recent work relating photosynthetic capacity to climate variables, environmental factors, and soil conditions shows promise for better capturing the dynamic nature of this parameter (e.g. Verheijen et al., 2013; Ali et al., 2015; Maire et al., 2015).


## 6. Conclusions

We evaluated the impacts on GPP, NEE, and NPP of new plant functional types in JULES. All changes were evaluated in version 4.2 with the canopy radiation model 5 option (Clark et al. 2011). At the base of the new PFTs was inclusion of new data from the TRY database. $N_m$

and LMA replaced the parameters $N_{l0}$ and $\sigma_l$. These were used to calculate new $V_{cmax,25}$, which was higher for all of the new PFTs compared to the original five, except for $C_3$ grasses.

The higher $V_{cmax,25}$ resulted in higher GPP. The GPP did not increase for $C_4$ grasses due to a lower quantum efficiency, or for cold grasslands due to a lower optimal temperature for $V_{cmax}$. Increases in NPP generally followed on from the increases in GPP.


A trade-off between LMA and leaf lifespan was enforced by changing parameters relating to leaf phenology, growth and senescence. The new parameter values changed the turnover rate of leaves on trees in the spring and fall, therefore altering the leaf lifespan in JULES in a manner consistent with observations. In JULES9$_{TRY}$, the median leaf lifespan of grasses and

shrubs were reduced, which improved the seasonal cycle at the relevant sites (Las Majadas, Tonzi, Fort Peck, Kaamanen, and Tomakai). The exception was the Bondville crop site.

Including the full range of updated parameters (in JULES9$_{ALL}$) resulted in an improved seasonal cycle of GPP at ten sites and reductions to daily RMSE at 11 sites (out of 13 sites

with GPP data) compared to JULES9$_{TRY}$. The annual GPP was within the range of the Fluxnet observations at every site except for one (Hyytiälä). On a biome-scale, we compared GPP to the MTE product of Jung et al. (2011) and NPP to the MODIS17 product. GPP was improved in JULES9 for all eight biomes evaluated, although for the tundra and desert/shrubland biome the GPP was much lower than the MTE value. The global NPP was

slightly higher than observed, but JULES9 was closer to MOD17 in most biomes – the exceptions being the tropical forests, savannahs, and extratropical mixed forests where JULES9 was too high. The biome-averaged NPP from JULES9 was within the range of MOD17 NPP for all biomes.

Overall, the simulation of gross and net productivity was improved with the 9 PFTs. The present study can be thought of as a "bottom-up" approach to improving JULES fluxes, with

new parameters being based on large observationally based datasets. The next step for improving PFTs in JULES is to evaluate the 9 PFTs when the dynamic vegetation is turned on. This will be addressed in a follow-up paper. A complimentary, "top-down" method for reducing uncertainty in JULES is to optimise PFT parameters based on minimising errors between simulated and observed fluxes. This is currently being done with adJULES, an adjoint version of JULES (Raoult et al., 2016). Future model development within JULES will have more flexibility for improving the model with more PFTs, and the improvements presented in this study increase our confidence in using JULES in carbon cycle studies.

**Code Availability**

The simulations discussed in this manuscript were done on JULES version 4.2. The exact code is available from Dropbox at:

https://www.dropbox.com/s/ydt4lxe1320fhty/JULES4.2_PFTs.tar.gz?dl=0, along with namelist files for the Fluxnet simulations.

**Acknowledgements**

We gratefully acknowledge all funding bodies. AH was funded by the NERC Joint Weather and Climate Research Programme and NERC grant NE/K016016/1. The study has been supported by the TRY initiative on plant traits (http://www.try-db.org). The TRY initiative and database is hosted, developed and maintained by J. Kattge and G. Bönisch (Max Planck Institute for Biogeochemistry, Jena, Germany). TRY is currently supported by DIVERSITAS/Future Earth and the German Centre for Integrative Biodiversity Research (iDiv) Halle-Jena-Leipzig. OA acknowledges the support of the Australian Research Council (CE140100008). Met Office authors were supported by the Joint DECC/Defra Met Office Hadley Centre Climate Programme (GA01101). VO was supported by RSF (RNF) (Project

14-50-00029). JP acknowledges support from the European Research Council Synergy grant ERC-SyG-2013-610028, IMBALANCE-P, and ÜN from the Advanced grant ERC-AdG-322603, SIP-VOL+. We also thank Andrew Hartley (UK Met Office), who processed the

ESA Land Cover data to the 5 and 9 PFTs, and Nicolas Viovy (IPSL-LSCE), who kindly provided the CRU-NCEP driving data.

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

 **Tables and Figure captions**

**Table 1**. Parameters used for the 5 PFT experiment (JULES5). The standard PFTs are: broadleaf trees (BT), needle-leaf trees (NT), $C_3$ grass, $C_4$ grass, and shrubs (SH). $N_m$ was calculated by dividing the default $N_{l0}$ by $C_{mass}$ (0.5 in this study), LMA was given the same value as $\sigma_L$, and $s_v$ was calculated to yield the same $V_{cmax,25}$ as with the default 5 PFTs. All other parameters were taken from Clark et al. (2011).

|  | BT | NT | $C_3$ | $C_4$ | SH |
|---|---|---|---|---|---|
| $a_{wl}$ | 0.65 | 0.65 | 0.005 | 0.005 | 0.10 |
| $D_{crit}$ | 0.09 | 0.06 | 0.10 | 0.075 | 0.10 |
| $d_T$ | 9 | 9 | 9 | 9 | 9 |
| $f_0$ | 0.875 | 0.875 | 0.900 | 0.800 | 0.900 |
| $f_d$ | 0.010 | 0.015 | 0.015 | 0.025 | 0.015 |
| $i_v$ | 0 | 0 | 0 | 0 | 0 |
| $L_{max}$ | 9 | 6 | 4 | 4 | 4 |
| $L_{min}$ | 1 | 1 | 1 | 1 | 1 |
| LMA | 0.075 | 0.200 | 0.050 | 0.100 | 0.100 |
| $N_a^{a}$ | 1.73 | 3.30 | 1.83 | 3.00 | 3.00 |
| $N_m$ | 0.023 | 0.0165 | 0.0365 | 0.030 | 0.030 |
| $rootd$ | 3 | 1 | 0.5 | 0.5 | 0.5 |
| $s_v$ | 21.33 | 8.00 | 32.00 | 8.00 | 16.00 |
| $T_{low}$ | 0 | -10 | 0 | 13 | 0 |
| $T_{off}$ | 5 | -40 | 5 | 5 | 5 |
| $T_{opt}$ | 32 | 22 | 32 | 41 | 32 |
| $T_{upp}$ | 36 | 26 | 36 | 45 | 36 |
| $V_{cmax,25}$ | 36.8 | 26.4 | 58.4 | 24.0 | 48.0 |
| $\alpha$ | 0.08 | 0.08 | 0.12 | 0.06 | 0.08 |
| $\gamma_0$ | 0.25 | 0.25 | 0.25 | 0.25 | 0.25 |
| $\gamma_p$ | 20 | 15 | 20 | 20 | 15 |
| $\mu_{rl}$ | 1.0 | 1.0 | 1.0 | 1.0 | 1.0 |
| $\mu_{sl}$ | 0.10 | 0.10 | 1.00 | 1.00 | 0.10 |

[a]These are derived from other parameters. Here $N_a$ is g N m$^{-2}$.

**Table 2**. Updated parameters used in JULES9$_{\text{ALL}}$. The new PFTs are: tropical broadleaf evergreen trees (BET-Tr), temperate broadleaf evergreen trees (BET-Te), needle-leaf evergreen trees (NET), needle-leaf deciduous trees (NDT), C$_3$ grass, C$_4$ grass, evergreen shrubs (ESH), and deciduous shrubs (DSH).

| | BET-Tr | BET-Te | BDT | NET | NDT | C$_3$ | C$_4$ | ESH | DSH |
|---|---|---|---|---|---|---|---|---|---|
| $a_{wl}$ | 0.65 | 0.65 | 0.65 | 0.65 | 0.75 | 0.005 | 0.005 | 0.10 | 0.10 |
| $D_{crit}$ | 0.090 | 0.090 | 0.090 | 0.060 | 0.041 | 0.051 | 0.075 | 0.037 | 0.030 |
| $d_T$ | 9 | 9 | 9 | 9 | 9 | 0 | 0 | 9 | 9 |
| $f_0$ | 0.875 | 0.892 | 0.875 | 0.875 | 0.936 | 0.931 | 0.800 | 0.950 | 0.950 |
| $f_d$ | 0.010 | 0.010 | 0.010 | 0.015 | 0.015 | 0.019 | 0.019 | 0.015 | 0.015 |
| $i_v$ | 7.21 | 3.90 | 5.73 | 6.32 | 6.32 | 6.42 | 0.00 | 14.71 | 14.71 |
| $L_{max}$ | 9 | 7 | 7 | 7 | 6 | 3 | 3 | 4 | 4 |
| $L_{min}$ | 1 | 1 | 1 | 1 | 1 | 1 | 1 | 1 | 1 |
| LMA | 0.1039 | 0.1403 | 0.0823 | 0.2263 | 0.1006 | 0.0495 | 0.1370 | 0.1515 | 0.0709 |
| $N_a{}^a$ | 1.76 | 2.02 | 1.74 | 2.61 | 1.87 | 1.19 | 1.55 | 2.04 | 1.54 |
| $N_m$ | 0.017 | 0.0144 | 0.021 | 0.0115 | 0.0186 | 0.0240 | 0.0113 | 0.0136 | 0.0218 |
| $rootd$ | 3 | 2 | 2 | 1.8 | 2 | 0.5 | 0.5 | 1 | 1 |
| $s_v$ | 19.22 | 28.40 | 29.81 | 18.15 | 23.79 | 40.96 | 20.48 | 23.15 | 23.15 |
| $T_{low}$ | 13 | 13 | 5 | 5 | -5 | 10 | 13 | 10 | 0 |
| $T_{off}$ | 0 | -40 | 5 | -40 | 5 | 5 | 5 | -40 | 5 |
| $T_{opt}$ | 39 | 39 | 39 | 33 | 34 | 28 | 41 | 32 | 32 |
| $T_{upp}$ | 43 | 43 | 43 | 37 | 36 | 32 | 45 | 36 | 36 |
| $V_{cmax,25}$ | 41.16 | 61.28 | 57.25 | 53.55 | 50.83 | 51.09 | 31.71 | 62.41 | 50.40 |
| $\alpha$ | 0.08 | 0.06 | 0.08 | 0.08 | 0.10 | 0.06 | 0.04 | 0.06 | 0.08 |
| $\gamma_0$ | 0.25 | 0.50 | 0.25 | 0.25 | 0.25 | 3.0 | 3.0 | 0.66 | 0.25 |
| $\gamma_p$ | 15 | 15 | 20 | 15 | 20 | 20 | 20 | 15 | 30 |
| $\mu_{rl}$ | 0.67 | 0.67 | 0.67 | 0.67 | 0.67 | 0.72 | 0.72 | 0.67 | 0.67 |
| $\mu_{sl}$ | 0.10 | 0.10 | 0.10 | 0.10 | 0.10 | 1.00 | 1.00 | 0.10 | 0.10 |

[a]These are derived from other parameters. Here $N_a$ is g N m$^{-2}$.

**Table 3.** Experiments for the Fluxnet site level evaluation.

| Experiment Number | Description |
|---|---|
| **0: JULES5** | 5 PFTs (Table 1) |
| 1 | 9 PFTs with $N_m$, LMA, and $V_{cmax,25}$ from TRY |
| **2: JULES9-TRY** | Exp. 1 + parameters affecting leaf lifespan |
| 3 | Exp. 2 + $f_0$ and $D_{crit}$ |
| 4 | Exp. 2 + $\alpha$ |
| 5 | Exp. 2 + adjusted $f_d$, $T_{upp}$, $T_{low}$, and $s_v$ |
| 6 | Exp. 2 + rootd, $a_{wl}$ |
| **7: JULES9** | All new PFT parameters (Table 2) |

**Table 4**. Sites used in the site simulations. Land cover is according to site PI.

| Site Name | Location | Simulated years | Land Cover | Dominant PFT(s) |
|---|---|---|---|---|
| **BR-Ma2** | Manaus, Brazil | 2002-2005 | Evergreen broadleaf forest | 100% BET |
| **BR-Sa1** | Santarem (Tapajós Forest, KM67), Brazil | 2002-2004 | Evergreen broadleaf forest | 100% BET |
| **BR-Sa3** | Santarem (Tapajós Forest, KM77), Brazil | 2001-2005 | Pasture | 20% BET, 75% $C_4$, 5% soil |
| **DE-Tha** | Tharandt, Germany | 1998-2006 | Needle-leaf evergreen forest | 100% NET |
| **ES-ES1** | El Saler, Spain | 1999-2006 | Needle-leaf evergreen forest | 100% NET |
| **ES-LMa** | Las Majadas, Spain | 2004-2006 | Closed shrub | 33% Temp-BET, 33% $C_3$, 33% ESh |
| **FI-Hyy** | Hyytiälä, Finland | 1998-2002 | Needle-leaf evergreen forest | 100% NET |
| **FI-Kaa** | Kaamanen, Finland | 2000-2005 | Wetland (simulated as $C_3$ grass) | 80% $C_3$ grass, 20% bare soil |
| **JP-Tom** | Tomakai, Japan | 2001-2003 | Needle-leaf deciduous plantation | 10% BDT, 10% NET, 80% NDT |
| **US-Bo1** | Bondville, Ill., US | 1997-2006 | Crop (rotating $C_3$/$C_4$) | 40% $C_3$, 40% $C_4$, 20% soil |
| **US-FPe** | Fort Peck, Mont., US | 2000-2006 | Grassland ($C_3$) | 80% $C_3$ grass, 20% bare soil |
| **US-Ha1** | Harvard, Mass., US | 1995-2001 | Broadleaf deciduous forest | 100% BDT |
| **US-MMS** | Morgan Monroe Forest, US | 2000-2004 | Broadleaf deciduous forest | 100% BDT |
| **US-Ton** | Tonzi, Calif., US | 2001-2006 | Woody savannah | 33% BDT, 33% $C_3$, 33% DSh |

**Table 5.** Comparison of simulated and observed annual GPP and NPP at Fluxnet sites, listed in order from most to least productive. Units: g C m$^{-2}$ yr$^{-1}$. Results are color-coded so blue shows when there is an improvement. The GPP and NPP are based on similar data processing between the Fluxnet observations and model. Sources: [1]Malhi 2009; [2]Gower and Richards, 1990, assuming 0.5gC/g biomass

| Site | GPP | | | NPP | | |
|---|---|---|---|---|---|---|
| | JULES5 | JULES9 | OBS | JULES5 | JULES9 | OBS |
| BR-Sa1 | 2671 | **2795** | 3314±600 | 850 | 1048 | 1440±130[1] |
| BR-Ma2 | 2848 | **3225** | 3285±835 | 867 | 1198 | 1011±140[1] |
| BR-Sa3 | 3318 | 2116 | | 1623 | 1125 | |
| DE-Tha | 1364 | **1876** | 1923±547 | 700 | 1004 | |
| JP-Tom | 1306 | **1361** | 1723±641 | 691 | 747 | 1100[2] |
| ES-ES1 | 1164 | 1087 | 1458±383 | 513 | 404 | |
| US-MMS | 1135 | **1234** | 1445±463 | 603 | 693 | |
| US-Ha1 | 1229 | **1438** | 1433±531 | 686 | 851 | |
| US-Bo1 | 896 | **1006** | 1233±568 | 457 | 591 | |
| ES-LMA | 1095 | 1257 | 1133±305 | 500 | 644 | |
| FI-Hyy | 1124 | 1465 | 1084±324 | 605 | 834 | |
| US-Ton | 818 | 794 | 924±256 | 365 | 405 | |
| US-FPe | 238 | **368** | 354±185 | 88 | 192 | |
| FI-Kaa | 633 | **512** | 297±126 | 359 | 311 | |


**Table 6a**. Area-weighted GPP from each biome (g C m$^{-2}$ yr$^{-1}$). The biome total GPP from MTE is given in Pg C yr$^{-1}$ to give perspective of each biome's role in the global total.

| Biome | JULES5 | JULES9 | JULES5-ALL | MTE | MTE total |
|---|---|---|---|---|---|
| Tropical forest | 2403±217 | 2295±191 | 2505±217 | 2244±297 | 49.9 |
| Tropical forest: Only BET-Tr. | 2924±144 | 2955±147 | 3279±178 | 2790±273 | |
| Tropical savannah | 1355±244 | 1268±223 | 1320±237 | 1111±257 | 21.9 |
| Extratropical mixed forests | 947±147 | 1082±158 | 1119±167 | 1119±212 | 2.9 (13.4*) |
| Boreal and coniferous forests | 514±99 | 597±118 | 645±122 | 650±203 | 12.1 |
| Temperate grasslands | 420±145 | 465±138 | 477±140 | 509±184 | 8.1 |
| Deserts and shrublands | 82±48 | 91±46 | 91±47 | 283±200 | 4.9 |
| Tundra | 86±20 | 94±20 | 101±20 | 279±233 | 1.9 |
| Mediterranean Woodlands | 324±147 | 407±136 | 405±140 | 510±190 | 1.5 |

*Value for EMF biome when agricultural mask is not applied.

**Table 6b**. Area-weighted NPP from each biome (g C m$^{-2}$ yr$^{-1}$).

| Biome | JULES5 | JULES9 | JULES5-ALL | MODIS17 |
|---|---|---|---|---|
| Tropical forest | 956±144 | 1007±125 | 951±143 | 786±352 |
| Only BET-Tr. | 1141±101 | 1233±103 | 1109±126 | 929±315 |
| Tropical savannah | 527±158 | 591±143 | 584±152 | 451±319 |
| Extratropical mixed forests | 586±93 | 631±104 | 640±110 | 563±231 |
| Boreal and coniferous forests | 307±65 | 358±77 | 385±80 | 350±155 |
| Temperate grasslands | 180±94 | 243±89 | 242±90 | 304±247 |
| Deserts and shrublands | 16±29 | 35±29 | 33±29 | 111±133 |
| Tundra | 52±14 | 61±13 | 65±13 | 136±94 |
| Mediterranean Woodlands | 118±94 | 201±89 | 195±89 | 324±184 |

**Table SM1.** List of parameters and symbols in the text.

| Symbol | Units | Equation | Description | Default Value[a] |
|---|---|---|---|---|
| $A_l$ | kg C m$^{-2}$ s$^{-1}$ | 5 | Leaf-level photosynthesis | |
| $a_{wl}$ | kg C m$^{-2}$ | 24 | Allometric coefficient | |
| $a_{ws}$ | -- | 24 | Ratio of total to respiring stem carbon | |
| $b_{wl}$ | -- | 24 | Allometric exponent | 1.667 |
| $C_i$ | Pa | 6 | Internal leaf $CO_2$ concentration | |
| $C_{mass}$ | kg C [kg biomass]$^{-1}$ | 23 | Leaf carbon concentration per unit mass | 0.5 for this study |
| $C_s$ | Pa | 6 | Leaf surface $CO_2$ concentration | |
| $D_{crit}$ | kg kg$^{-1}$ | 7 | Critical humidity deficit | |
| $d_T$ | -- | 16 | Rate of change of leaf turnover with temperature | |
| $f_0$ | -- | 7 | Stomatal conductance parameter | |
| $f_d$ | -- | 4 | Leaf dark respiration coefficient | |
| $g_s$ | m s$^{-1}$ | 6 | Leaf-level stomatal conductance | |
| $i_v$ | µmol $CO_2$ m$^{-2}$ s$^{-1}$ | 19 | Intercept for relationship between $N_A$ and $V_{cmax,25}$ | |
| $k_n$ | -- | 3, 20 | Extinction coefficient for nitrogen | 0.78 |
| $h$ | m | 13, 23, 24 | Canopy height | |
| $L_{bal}$ | m$^2$ m$^{-2}$ | 12, 13, 22-24 | Balanced leaf area index (maximum LAI given the plant's height) | |
| $L_{max}$ | m$^2$ m$^{-2}$ | | Maximum LAI | |
| $L_{min}$ | m$^2$ m$^{-2}$ | | Minimum LAI | |
| LMA | kg m$^{-2}$ | 18, 21, 22 | Leaf mass per unit area (new parameter) | |
| $N_a$ | kg N m$^{-2}$ | 18 | Leaf nitrogen per unit area | |
| $n_{eff}$ | mol $CO_2$ m$^{-2}$ s$^{-1}$ kg C [kg N]$^{-1}$ | 3 | Constant relating leaf nitrogen to Rubisco carboxylation capacity | |
| $N_{l0}$ | kg N [kg C]$^{-1}$ | 3 | Top leaf nitrogen concentration (old parameter, mass basis) | |
| $N_m$ | kg N kg$^{-1}$ | 18, 21-23 | Top leaf nitrogen concentration (new parameter) | |
| $N_l$ | kg N m$^{-2}$ | 11, 21 | Total leaf nitrogen concentration | |
| $N_r$ | kg N m$^{-2}$ | 12, 22 | Total root nitrogen concentration | |
| $N_s$ | kg N m$^{-2}$ | 13, 23 | Total stem nitrogen concentration | |
| $p$ | -- | 17 | Phenological state (LAI/$L_{bal}$) | |
| $Q_{10,leaf}$ | -- | 2 | Constant for exponential term in temperature function of $V_{cmax}$ | 2 |
| $R_a$ | kg C m$^{-2}$ s$^{-1}$ | 8 | Total plant autotrophic respiration | |
| $R_d$ | kg C m$^{-2}$ s$^{-1}$ | 4, 5 | Leaf dark respiraiton | |
| $r_g$ | -- | 10 | Growth respiration coefficient | 0.25 |
| $rootd$ | m | | e-folding root depth | |
| $s_v$ | µmol $CO_2$ g N$^{-1}$ s$^{-1}$ | 19 | Slope between $N_A$ and $V_{cmax,25}$ | |
| $T_{low}$ | °C | 1 | Upper temperature parameter for $V_{cmax}$ | |
| $T_{off}$ | °C | 16 | Threshold temperature for phenology | |
| $T_{opt}$[b] | °C | | Optimal temperature for $V_{cmax}$ | |
| $T_{upp}$ | °C | 1 | Upper temperature parameter for $V_{cmax}$ | |
| $V_{cmax,25}$ | µmol m$^{-2}$ s$^{-1}$ | 1, 9 | The maximum rate of carboxylation of | |

| | | | Rubisco at 25°C | |
|---|---|---|---|---|
| $W$ | kg C m$^{-2}$ s$^{-1}$ | 5 | Smoothed minimum of the potential limiting rates of phososynthesis | |
| $\alpha$ | mol $CO_2$ [mol PAR photons]$^{-1}$ | | Quantum efficiency | |
| $\beta$ | -- | 5 | Soil moisture stress factor | |
| $\Gamma^*$ | Pa | 7 | $CO_2$ compensation point | |
| $\gamma_0$ | [360 days]$^{-1}$ | 16 | Minimum leaf turnover rate | |
| $\gamma_{lm}$ | [360 days]$^{-1}$ | 16 | Leaf turnover rate | |
| $\gamma_p$ | [360 days]$^{-1}$ | 17 | Leaf growth rate | 20 |
| $\mu_{rl}$ | -- | 12, 22 | Ratio of nitrogen concentration in roots and leaves | |
| $\mu_{sl}$ | -- | 13, 23 | Ratio of nitrogen concentration in stems and leaves | |
| $\eta_{sl}$ | kg C m$^{-2}$ LAI$^{-1}$ | 13, 23 | Live stemwood coefficient | 0.01 |
| $\sigma_L$ | kg C m$^{-2}$ LAI$^{-1}$ | 11, 12 | Specific leaf density (old parameter) | |

[a]Default values only provided for non-PFT-dependent parameters.


**Table SM2**. New trait-based parameters for 5 PFTs that are consistent with TRY data.

| | BT | NT | C3 | C4 | SH |
|---|---|---|---|---|---|
| $N_m$ | 0.0185 | 0.0117 | 0.0240 | 0.0113 | 0.0175 |
| LMA | 0.1012 | 0.2240 | 0.0495 | 0.1370 | 0.1023 |
| $s_v$ | 25.48 | 18.15 | 40.96 | 20.48 | 23.15 |
| $i_v$ | 6.12 | 6.32 | 6.42 | 0.00 | 14.71 |
| $V_{cmax,25}$ | 53.84 | 53.88 | 55.08 | 31.71 | 56.15 |
| $T_{off}$ | 5 | -40 | 5 | 5 | -40 |
| $d_T$ | 9 | 9 | 0 | 0 | 9 |
| $\gamma_0$ | 0.25 | 0.25 | 3.0 | 3.0 | 0.66 |
| $\gamma_p$ | 20 | 15 | 20 | 20 | 15 |
| $L_{min}$ | 1 | 1 | 1 | 1 | 1 |
| $L_{max}$ | 9 | 7 | 3 | 3 | 4 |
| $D_{crit}$ | 0.09 | 0.06 | 0.051 | 0.075 | 0.037 |
| $f_0$ | 0.875 | 0.875 | 0.931 | 0.800 | 0.950 |
| $f_d$ | 0.010 | 0.015 | 0.019 | 0.019 | 0.015 |
| *rootd* | 3 | 2 | 0.5 | 0.5 | 1 |
| $T_{low}$ | 5 | 0 | 10 | 13 | 0 |
| $T_{opt}$ | 39 | 32 | 28 | 41 | 32 |
| $T_{upp}$ | 43 | 36 | 32 | 45 | 36 |
| $\alpha$ | 0.08 | 0.08 | 0.06 | 0.04 | 0.08 |
| $\mu_{rl}$ | 0.67 | 0.67 | 0.72 | 0.72 | 0.67 |

**Figure 1.** Trade offs between leaf mass per unit area (LMA; kg m$^{-2}$) and (a,c) leaf nitrogen (g

g$^{-1}$), and between LMA and (b,d) leaf lifespan (LL). (a,b) Parameters in the standard JULES,

converted from $N_{l0}$ and $\sigma_l$ based on 0.4 kg C per kg dry mass (assumed parameter in JULES

from Clark et al., 2011). (c,d) Median values from the TRY database for the new 9 PFTs. In

(b) and (d), the filled circles show the observed data and the open shapes show the median

values from global simulations of JULES from 1982-2012. Vertical and horizontal lines show

the range of vales between the lower and upper quartile of data.

**Figure 2.** $V_{\text{cmax},25}$ for the new nine PFTs (black), from the comparable PFT from the TRY

data (Kattge et al., 2009) (green), and from the standard 5 PFTs (red). Asterisks indicate the

$V_{\text{cmax},25}$ for JULES9 prior to calibration based on the Fluxnet sites. The standard deviation

reported in Kattge et al. (2009) are also shown for the observations with the vertical lines.

BET-Tr=Tropical broadleaf evergreen trees, BET-Te=Temperate broadleaf evergreen trees,

BDT=Broadleaf deciduous trees, NET=Needle-leaf evergreen trees, NDT=Needle-leaf

deciduous trees, C3G= $C_3$ grass, C4G= $C_4$ grass, ESh=Evergreen shrubs, DSh=Deciduous

shrubs.

**Figure 3.** (a) Dominant vegetation type from the ESA LC_CCI data set, aggregated to the

new 9 PFTs. (b) Color-coded map of global biomes, based on World Wildlife Fund biomes.

**Figure 4**. Relative changes in daily RMSE (Eq. 24) and monthly correlation coefficients (Eq.

25) for the JULES experiments in Table 4 compared to JULES5. Yellows and reds indicate

an improvement in JULES compared to the Fluxnet observations.

**Figure 5a.** Monthly mean fluxes of GPP. Observations ± standard deviation from Fluxnet are shown with triangles and vertical lines. The three JULES simulations are: JULES5 with standard 5 PFTs (JULES5, red); JULES with 9 PFTs and new LMA, $N_m$, and $V_{cmax,25}$ from TRY (JULES9$_{TRY}$, orange); JULES9-TRY plus new parameters for the PFTs as discussed in Section 2.3 (JULES9$_{ALL}$, blue). Also shown are the daily root mean square error (rmse) based on daily fluxes and the correlation coefficient (r) based on monthly mean fluxes for all years of the simulations. Site information is given in Table 3. All units are in g C m$^{-2}$ d$^{-1}$.

**Figure 5b.** As in 5a but for monthly anomalies of NEE.

**Figure 6.** Annual GPP and NPP for the eight biomes shown in Fig. 3b. Biome abbreviations are: D=Deserts, M=Mediterranean woodlands, TU=Tundra, TG=Temperate grasslands, TS=Tropical savannahs, BCF=boreal and coniferous forests, EMF=Extra-tropical mixed forests, TF=Tropical forests.

**Figure 7.** Area-averaged seasonal cylces of GPP from the biomes shown in Fig. 3b, comparing JULES5, JULES9, and the Jung et al. (2011) MTE. Also shown are the temperature and precipitation from the CRU-NCEP dataset used to force the JULES simulations. The gray shading in the GPP plots shows the MTE GPP ±1 standard deviation based on the area-averaged standard deviations of monthly fluxes for each grid cell.

**Figure 8.** Global maps of carbon cycle fluxes from 2000-2012. The observation sources are: MTE (GPP) and MODIS MOD17 (NPP, 2000-2013).

**Figure 9.** Differences between modelled and observed GPP (observed = MTE) and NPP (observed=MOD17). a,b) JULES with the standard 5 PFTs and default parameters; c,d) JULES with 5 PFTs and improved parameters; e,f) JULES with 9 PFTs and improved parameters.

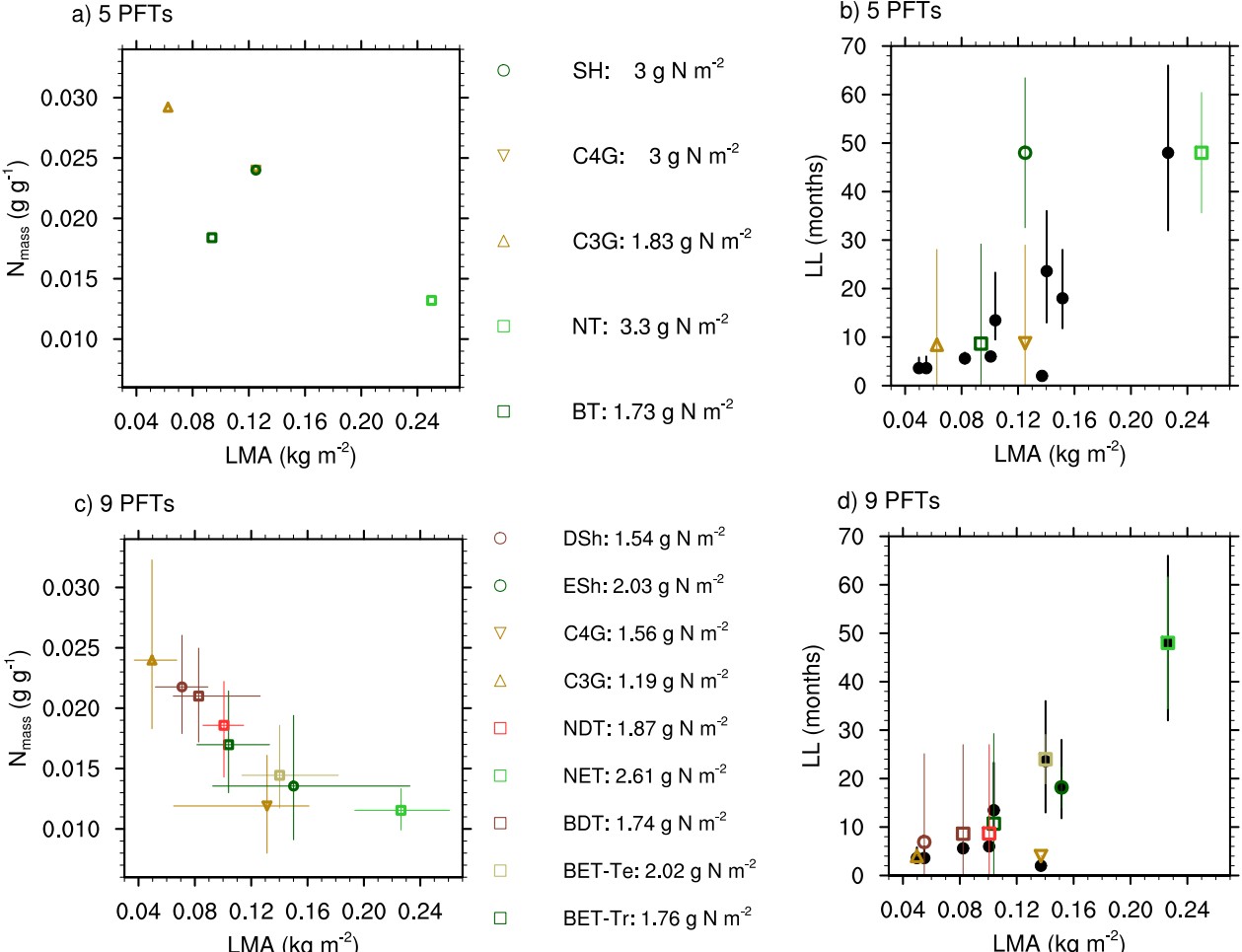

**Figure 1.** Trade offs between leaf mass per unit area (LMA; kg m$^{-2}$) and (a,c) leaf nitrogen (g g$^{-1}$), and between LMA and (b,d) leaf lifespan (LL). (a,b) Parameters in the standard JULES, converted from $N_{l0}$ and $\sigma_l$ based on 0.4 kg C per kg dry mass (assumed parameter in JULES from Clark et al., 2011). (c,d) Median values from the TRY database for the new 9 PFTs. In (b) and (d), the filled circles show the observed data and the open shapes show the median values from global simulations of JULES from 1982-2012. Vertical and horizontal lines show the range of vales between the lower and upper quartile of data.

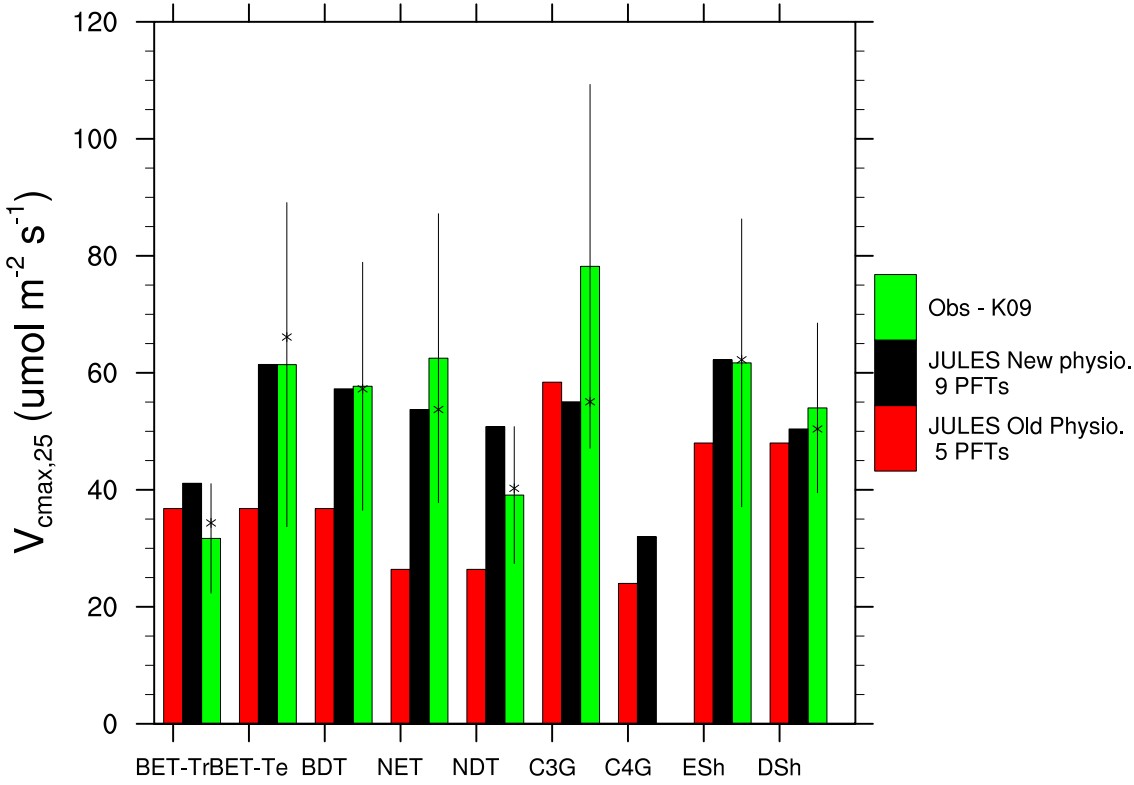

**Figure 2.** $V_{cmax,25}$ for the new nine PFTs (black), from the comparable PFT from the TRY data (Kattge et al., 2009) (green), and from the standard 5 PFTs (red). Asterisks indicate the $V_{cmax,25}$ for JULES9 prior to tuning based on the Fluxnet sites. The standard deviation reported in Kattge et al. (2009) are also shown for the observations with the vertical lines. BET-Tr=Tropical broadleaf evergreen trees, BET-Te=Temperate broadleaf evergreen trees, BDT=Broadleaf deciduous trees, NET=Needleleaf evergreen trees, NDT=Needleleaf deciduous trees, C3G=C3 grass, C4G=C4 grass, ESh=Evergreen shrubs, DSh=Deciduous shrubs.

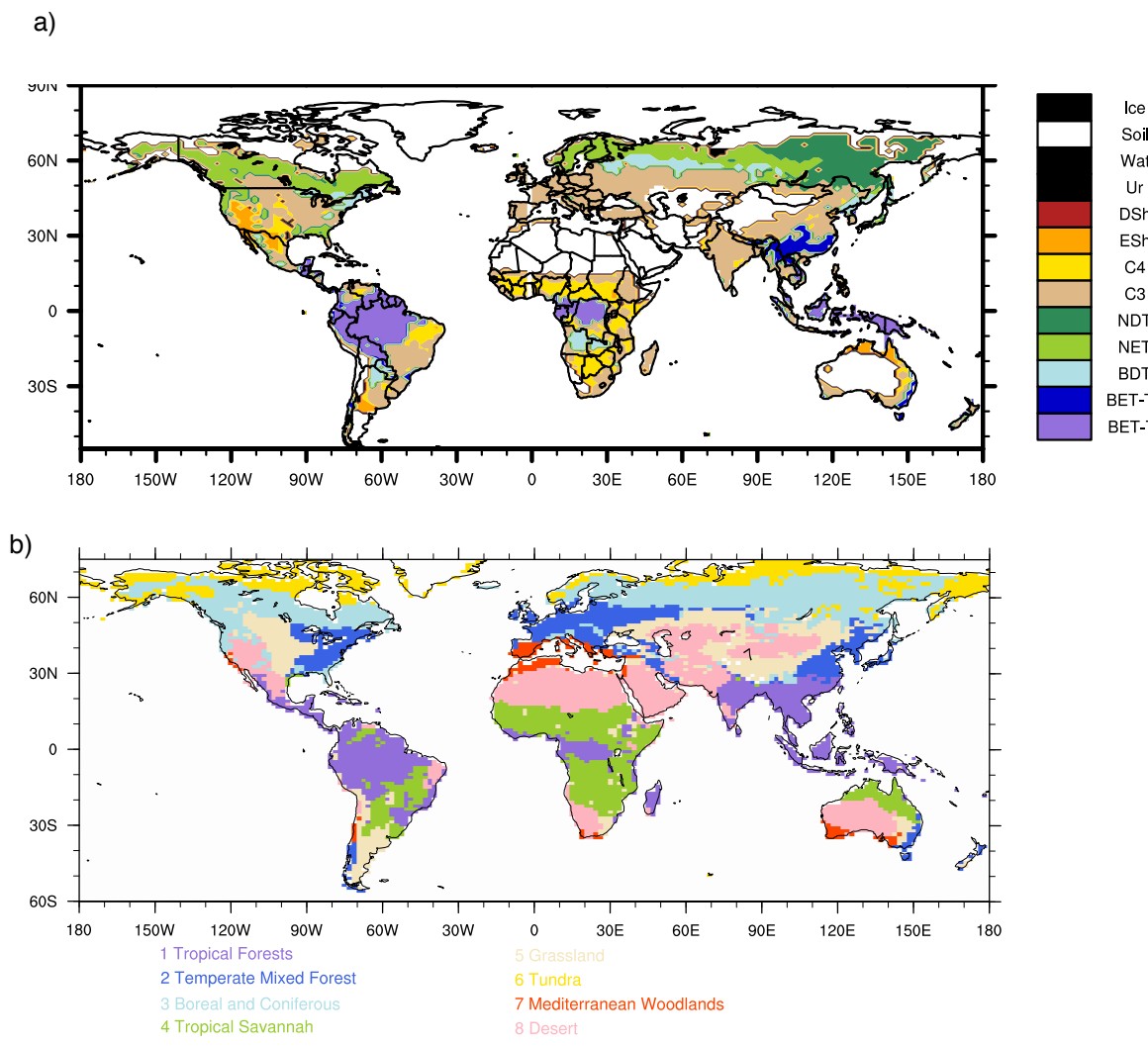

a)

b)

1 Tropical Forests
2 Temperate Mixed Forest
3 Boreal and Coniferous
4 Tropical Savannah

5 Grassland
6 Tundra
7 Mediterranean Woodlands
8 Desert

**Figure 3.** (a) Dominant vegetation type from the ESA LC_CCI data set, aggregated to the new 9 PFTs. (b) Color-coded map of global biomes, based on World Wildlife Fund biomes.

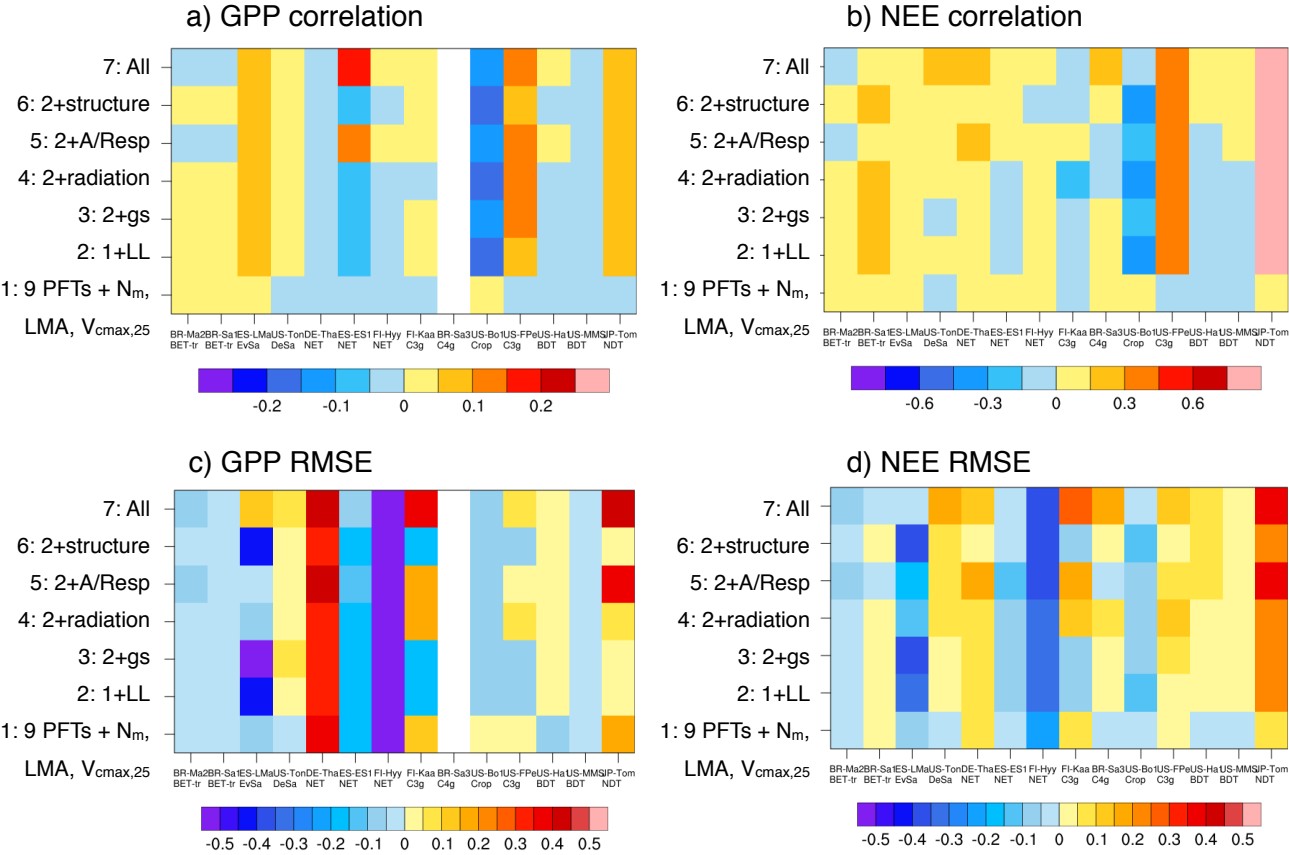

Figure 4. Relative changes in daily RMSE (Eq. 24) and monthly correlation coefficients (Eq. 25) for the JULES experiments in Table 4 compared to JULES5. Yellows and reds indicate an improvement in JULES compared to the Fluxnet observations.

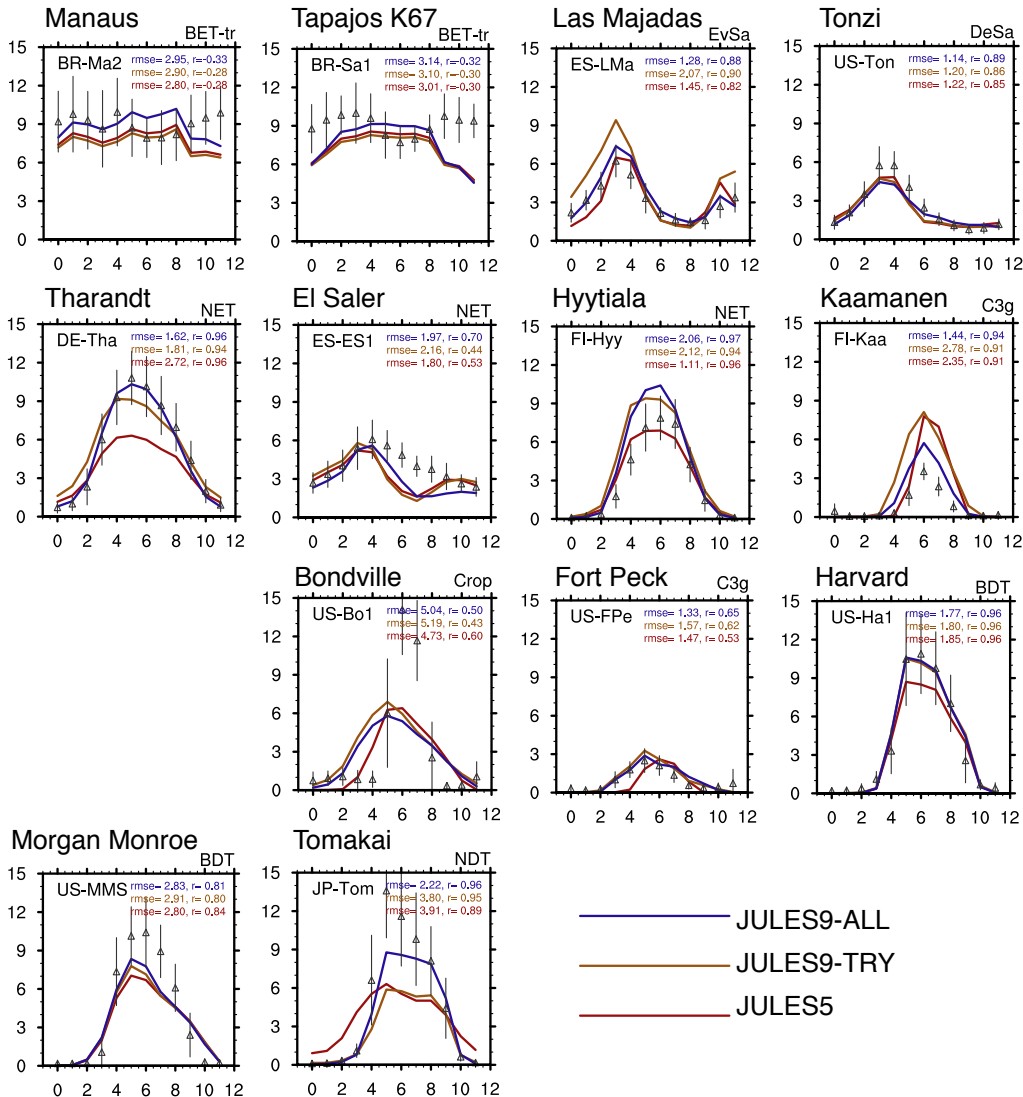

**Figure 5a.** Monthly mean fluxes of GPP. Observations ± standard deviation from Fluxnet are shown with triangles and vertical lines. The three JULES simulations are: JULES5 with standard 5 PFTs (JULES5, red); JULES with 9 PFTs and new LMA, $N_m$, and $V_{cmax,25}$ from TRY (JULES9-TRY, orange); JULES9-TRY plus new parameters for the PFTs as discussed in Section 2.3 (JULES9-ALL, blue). Also shown are the daily root mean square error (rmse) based on daily fluxes and the correlation coefficient (r) based on monthly mean fluxes for all years of the simulations. Site information is given in Table 3. All units are in g C m$^{-2}$ d$^{-1}$.

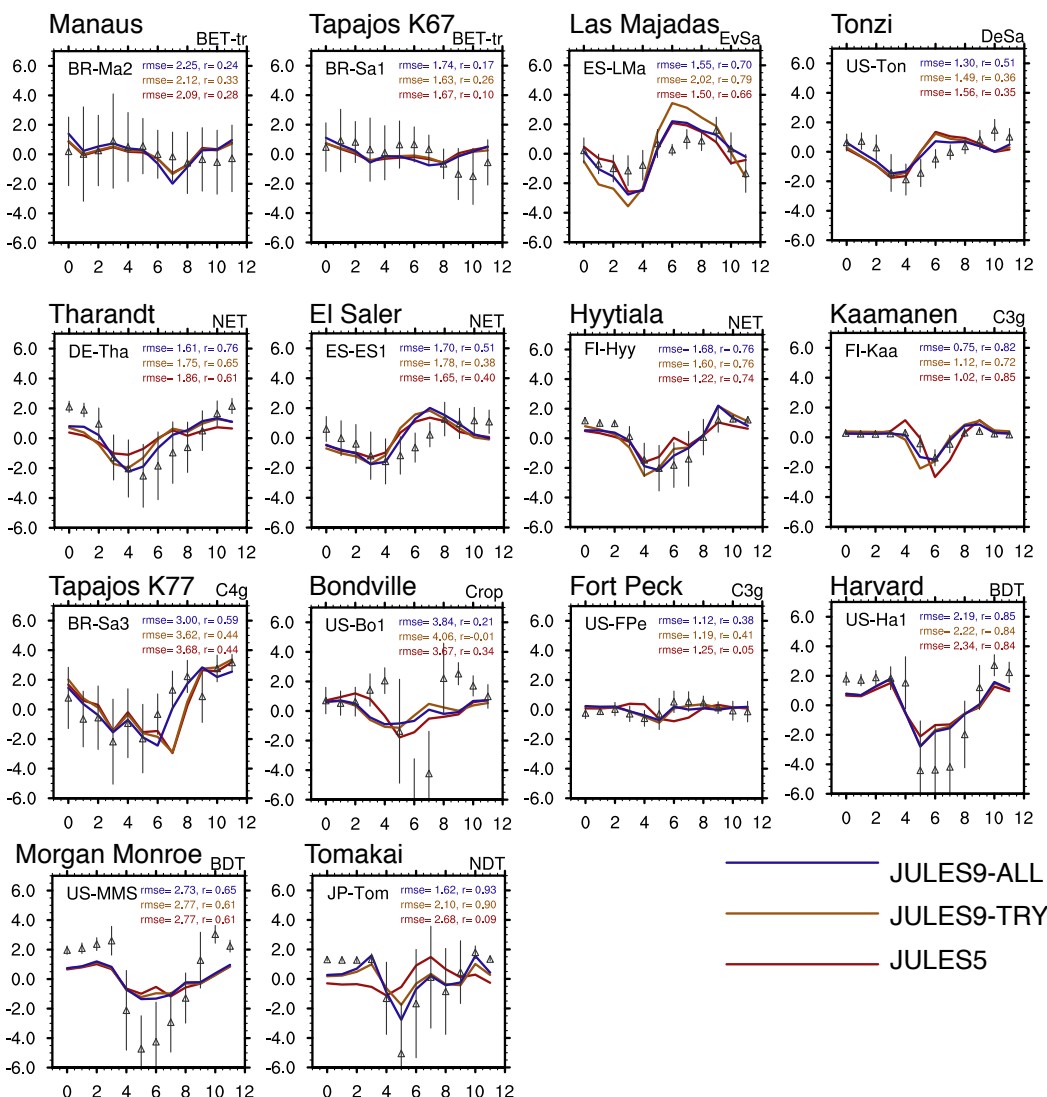

**Figure 5b.** As in 5a but for monthly anomalies of NEE.

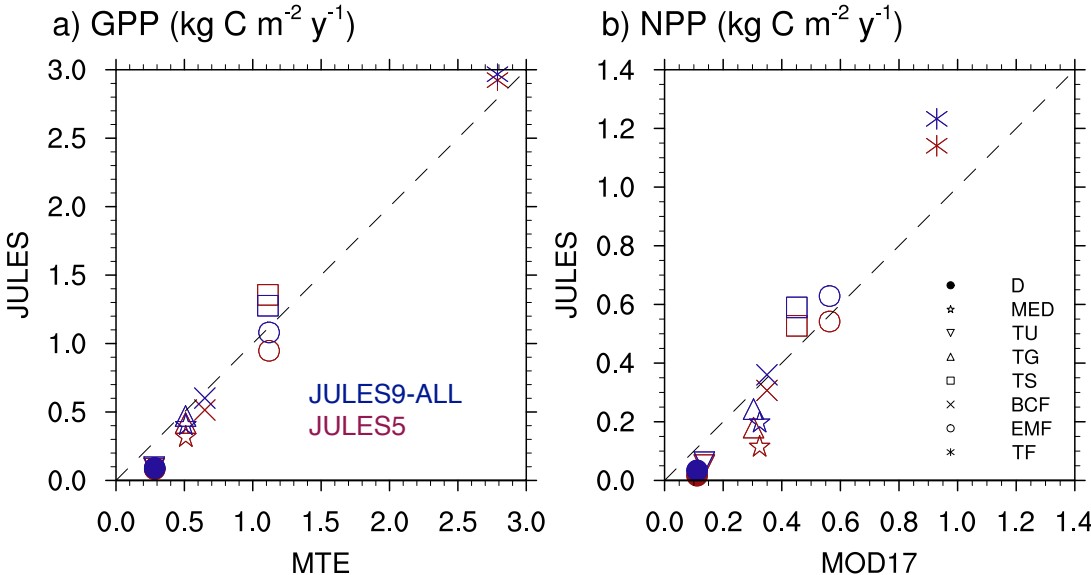

**Figure 6.** Annual GPP and NPP for the eight biomes shown in Fig. 3b. Biome abbreviations are: D=Deserts, M=Mediterranean woodlands, TU=Tundra, TG=Temperate grasslands, TS=Tropical savannahs, BCF=boreal and coniferous forests, EMF=Extra-tropical mixed forests, TF=Tropical forests.

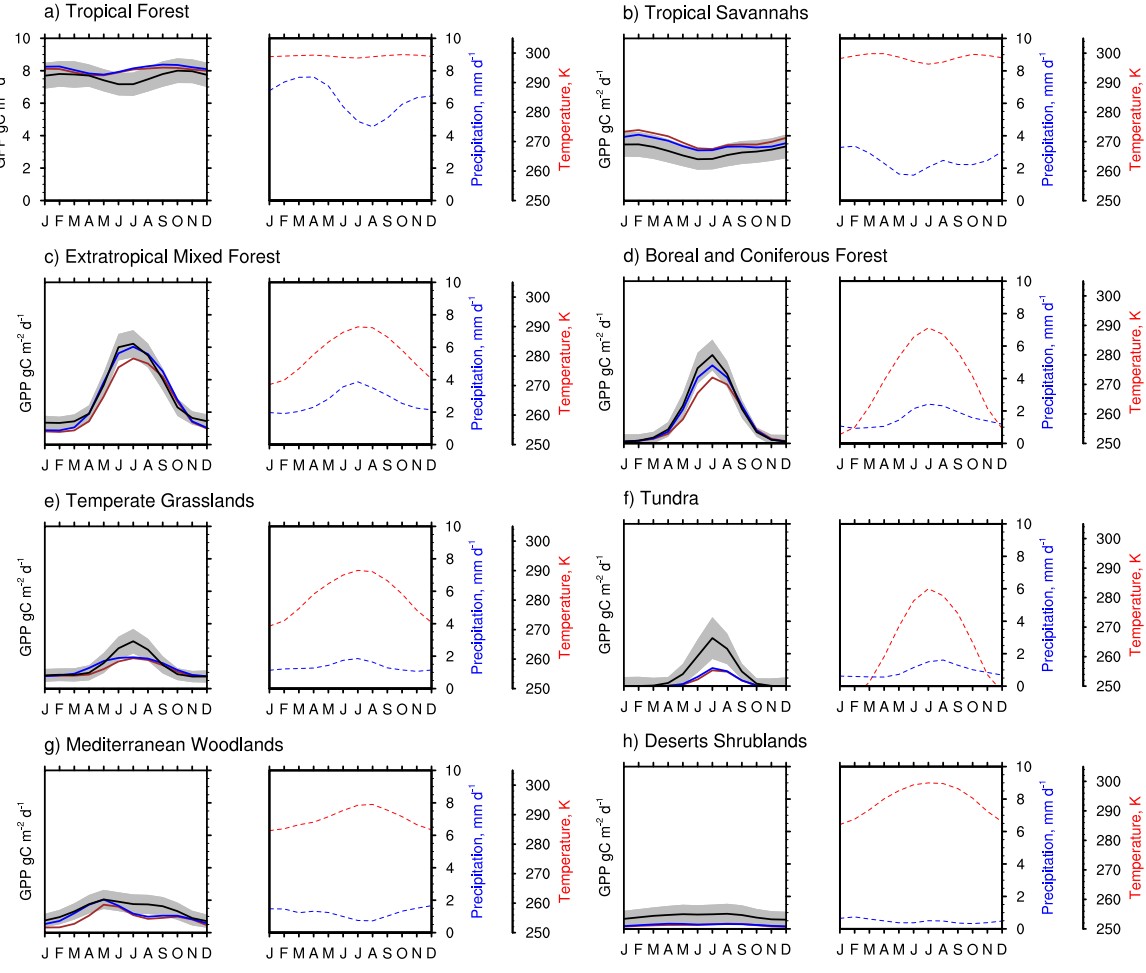

**Figure 7.** Area-averaged seasonal cylces of GPP from the biomes shown in Fig. 3, comparing JULES5, JULES9, and the Jung et al. (2011) MTE. Also shown are the temperature and precipitation from the CRU-NCEP dataset used to force the JULES simulations. The gray shading in the GPP plots shows the MTE GPP ±1 standard deviation based on the area-averaged standard deviations of monthly fluxes for each grid cell.

—— MTE

—— JULES9-ALL

—— JULES5

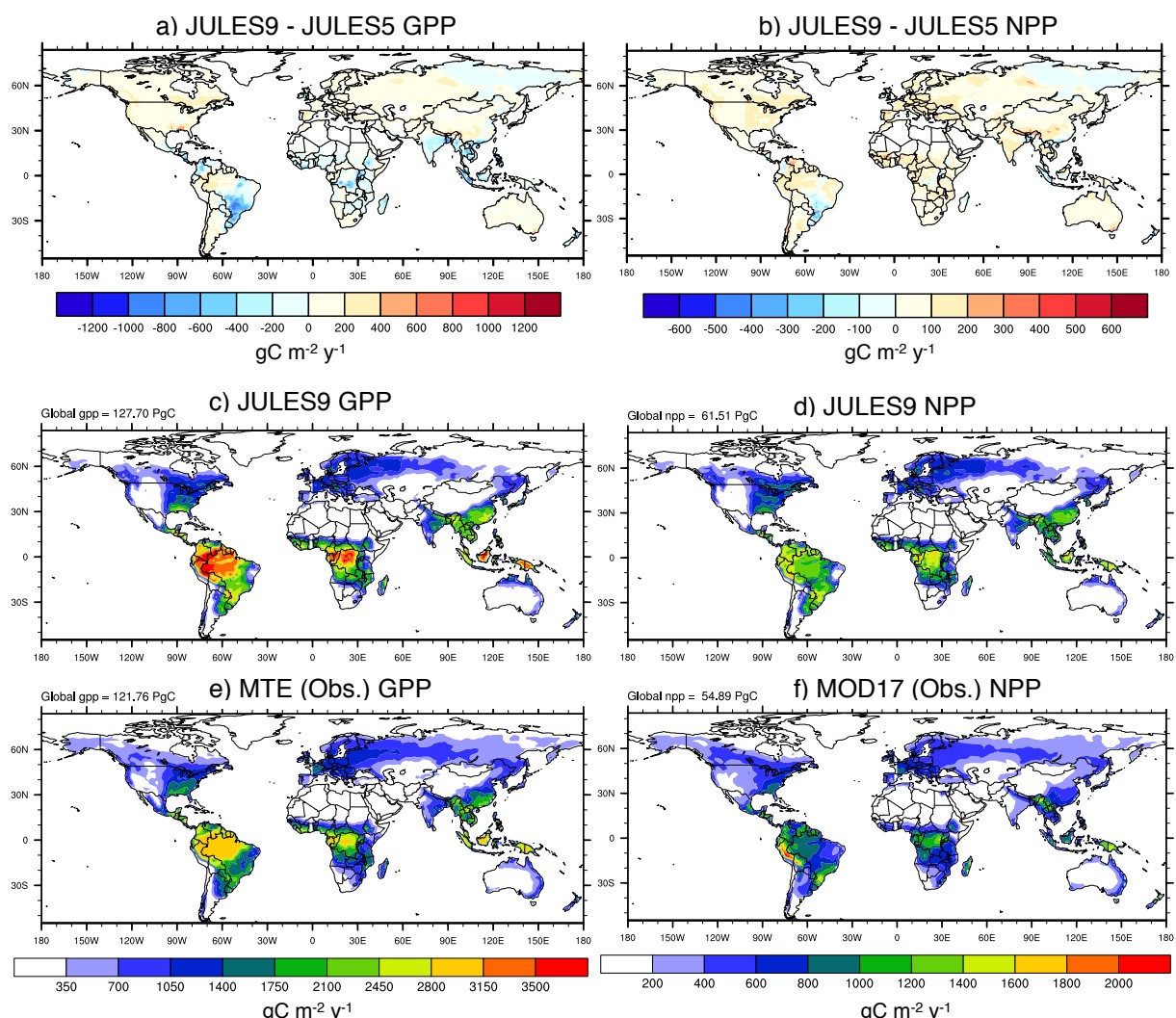

**Figure 8.** Global maps of carbon cycle fluxes from 2000-2012. The observation sources are: MTE (GPP), and MODIS MOD17 (NPP, 2000-2013).

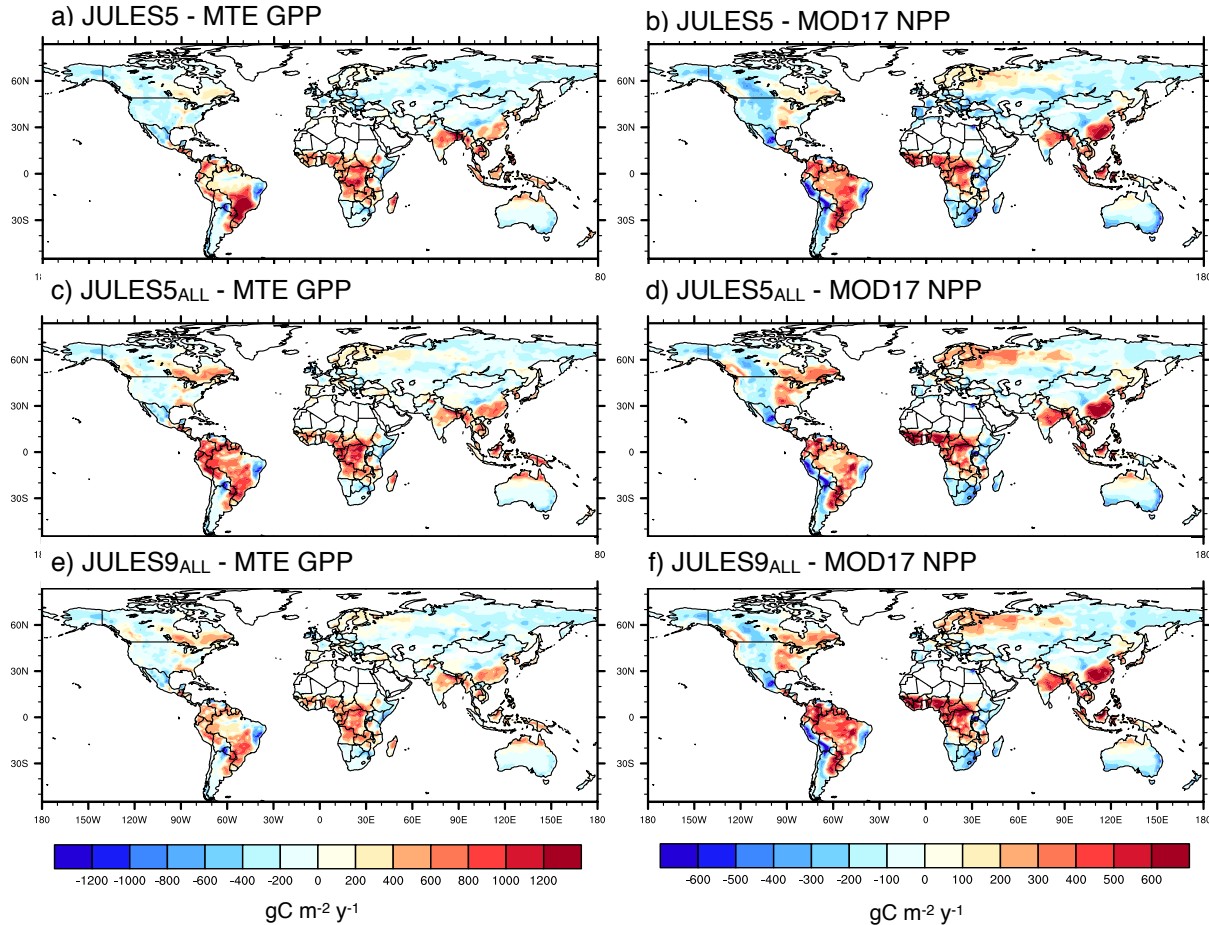

**Figure 9.** Differences between modelled and observed GPP (observed = MTE) and NPP (observed=MOD17). a,b) JULES with the standard 5 PFTs and default parameters; c,d) JULES with 5 PFTs and improved parameters; e,f) JULES with 9 PFTs and improved parameters.