# Peer review of "Improved representation of plant functional types and physiology in the Joint UK Land Environment Simulator (JULES v4.2) using plant trait information"

_Geoscientific Model Development, 2016_

## Referee Comment (RC1) · Anonymous Referee #1 · 7 Mar 2016

The paper "Improved representation of plant functional types and physiology in the Joint UK Land Environment Simulator (JULES v4.2) using plant trait information" present several improvements of the JULES DGVM. these improvement are first based on increasing the number of PFTs from 5 to 9 to better represent the different types of leaves in the leaf economic spectrum including deciduous and evergreen trees and a separation between climate zones. Second improvement was done in estimation of leaf photosynthesis from leaf nitrogen and improvement of phenology considering a more realistic leaf longevity.

[Figure]

This is an important paper that allow to follow recent developments of the JULES model and perfectly fit to the objective GMD. The changes are sufficiently important to justify the publication of a paper. The paper is well written with a convincing evaluation of new model performances both at site level and at global scale. The results show a clear improvement of the model at different scales. For all these reasons I recommend the paper for publication. Here after are just some minor comments that could help to improve the manuscript:

- There is no real justification of the choice of 9 PFTs except as a minimum to represent the main leaves forms. Obviously, for technical reasons, the number of PFTs cannot be increase indefinitely and then a compromise should be find but it would be interesting to see if including a higher number of PFT should also give higher performances ? One way could be to look to the differences between simulated GPP and NPP and respectively Jung and MODIS maps for each pixels and each PFT. Then we could see if there is spatially coherent systematic bias that could show possible new PFT separation.

minor comments on figures:

- Figure 7: what represent the grey zone ?

- Figure 8: The figure is difficult to read mainly because this is tiny figures. Should it be possible to split it to have larger figures ?
* * *

---

## Referee Comment (RC2) · Anonymous Referee #2 · 8 Mar 2016

The authors present a version of the JULES land surface model with a more detailed dynamic vegetation model and show that this gives more accurate carbon fluxes than the traditional version of JULES. It is of great interest and should be published.

My only question is whether you could you have got the same answer by tuning the old version of JULES? Adding extra PFTs will cause greater complication than tuning parameters, especially when competition between PFTs is turned on. You say you corrected known biases in the model. Did these same biases get corrected in the original, 5 PFT version, or just the new version? If not, I think you should have added

an extra experiment to assess the relative impact on the flux from adding the additional PFTs and the tuning. Would just correcting the 5 PFT JULES have had the same impact as adding extra PFTs? I think that some discussion of this, and ideally an extra experiment, is needed.

Experiments 4+ are discussed before experiments 1 to 3 in the text. It would be easier to follow if all the experiments were described in the same way and in the same order. Perhaps move the method around line 515 from the results section to before the first mention of experiment 4?

Table SM 2 gives tuned parameters for the tuned 5 PFT JULES, but I cannot find a reference to that in the text. Is there a missing section?

"and updated the model phenology to include a trade-off between leaf lifespan and leaf mass per unit area." - Does your improvement not just change the leaf turnover rate and its impact on the carbon flux rather than the phenology, which is still controlled in the same way as traditional JULES?

---

## Author Comment (AC1) · 26 Apr 2016

Responses to Anonymous Referee #1 on submission to Geoscientific Model Development Discussion (doi:10.5194/gmd-2016-22).

Submitted by Anna Harper on behalf of myself and my co-authors.

We thank the referee for your helpful comments on the manuscript and for taking the time to review it. Below we include the referee comments in black and our responses in red. The supplement contains a revised manuscript with red indicating changed sections. All line numbers refer to that version of the manuscript. Note the revised manuscript also includes some edits of minor errors (all in red for traceability).

The paper "Improved representation of plant functional types and physiology in the Joint UK Land Environment Simulator (JULES v4.2) using plant trait information" present several improvements of the JULES DGVM. these improvement are first based on increasing the number of PFTs from 5 to 9 to better represent the different types of leaves in the leaf economic spectrum including deciduous and evergreen trees and a separation between climate zones. Second improvement was done in estimation of leaf photosynthesis from leaf nitrogen and improvement of phenology considering a more realistic leaf longevity.

This is an important paper that allow to follow recent developments of the JULES model and perfectly fit to the objective GMD. The changes are sufficiently important to justify the publication of a paper. The paper is well written with a convincing evaluation of new model performances both at site level and at global scale. The results show a clear improvement of the model at different scales. For all these reasons I recommend the paper for publication. Here after are just some minor comments that could help to improve the manuscript:

- There is no real justification of the choice of 9 PFTs except as a minimum to represent the main leaves forms. Obviously, for technical reasons, the number of PFTs cannot be increase indefinitely and then a compromise should be find but it would be interesting to see if including a higher number of PFT should also give higher performances?
One way could be to look to the differences between simulated GPP and NPP and respectively Jung and MODIS maps for each pixels and each PFT. Then we could see if there is spatially coherent systematic bias that could show possible new PFT separation.

It's true that the choice of PFTs is subjective. The 9 PFTs were chosen as they represent the range of deciduous and evergreen plant types with minimal externally determined bioclimatic limits. The distinction between tropical and temperate broadleaf evergreen trees exists to account for the important differences between these types of trees, as described in the Introduction. In particular, measured $V_{cmax}$ for a given leaf N per unit area ($N_A$) can be lower in tropical evergreen trees than in temperate broadleaf evergreen trees (Kattge et al., 2009), resulting in lower $V_{cmax}$ and maximum assimilation rates for tropical forests. We have added these justifications in the Discussion (Lines 759-763).

Previously JULES was hard-wired for 5 PFTs. An important step in going from 5 to 9 PFTs was removing this hardwiring. Now users can define the number of PFTs, so the 9 documented in this paper are a recommendation but can be adjusted in the future. This is now mentioned at Line 754-756.

However, it is a good suggestion to evaluate in a more objective way if an appropriate number of PFTs has been chosen. One logical way to further subdivide the PFTs is based on the biome maps, which is similar to the reviewer's suggestion. The analysis in the manuscript was based on an

original data set of 14 biomes, where some biomes were combined for a total of 7 biomes. In a new figure (Fig. SM6) we show the biases in JULES5 and JULES9 for 11 of the original 14 biomes (3 biomes are very small and had no visible differences in the maps: Tropical Coniferous forests, Mangroves, and Flooded grasslands/savannas). The area-weighted RMSE is given in the top left of each map. Some biomes do not show an improvement in JULES9 and this gives some indication where extra PFTs might improve the simulation: for example the Boreal Forests/Tiaga; Tundra; Mediterranean woodlands; and Desert/Xeric Shrublands. Also the biases are still very high for the tropical/subtropical forest and grassland biomes. These regions broadly agree with what was mentioned in Section 5.2. We have added a more specific recommendation for development in these regions at Line 770-772.

However we also caution against defining too many PFTs, as there is already overlap between the $N_m$ and LMA traits (Fig. 1c). In developing new PFTs it would be ideal to determine definitions that result in distinctive sets of traits. Future work will address the possibility of more PFTs or improved processes with the new framework for flexible PFTs in JULES.

minor comments on figures:
- Figure 7: what represent the grey zone ?
This is the standard deviation of the observed fluxes, based on the monthly means from all months. This information has been added to the caption.
- Figure 8: The figure is difficult to read mainly because this is tiny figures. Should it be possible to split it to have larger figures ?
We rearranged this figure so the individual panels can be larger.

Responses to Anonymous Referee #2 on submission to Geoscientific Model Development Discussion (doi:10.5194/gmd-2016-22).

Submitted by Anna Harper on behalf of myself and my co-authors.

We thank the referee for your helpful comments on the manuscript and for taking the time to review it. Below we include the referee comments in black (with the specific questions addressed in bold), and our responses in red. The supplement contains a revised manuscript with red indicating changed sections. All line numbers refer to that version of the manuscript. Note the revised manuscript also includes some edits of minor errors (all in red for traceability).

The authors present a version of the JULES land surface model with a more detailed dynamic vegetation model and show that this gives more accurate carbon fluxes than the traditional version of JULES. It is of great interest and should be published. **My only question is whether you could you have got the same answer by tuning the old version of JULES?** Adding extra PFTs will cause greater complication than tuning parameters, especially when competition between PFTs is turned on. You say you corrected known biases in the model. **Did these same biases get corrected in the original, 5 PFT version, or just the new version?** If not, I think you should have added an extra experiment to assess the relative impact on the flux from adding the additional PFTs and the tuning. **Would just correcting the 5 PFT JULES have had the same impact as adding extra PFTs?** I think that some discussion of this, and ideally an extra experiment, is needed.

First, we address the question of tuning. In this study, we have used observations to constrain the model. This has improved the model and it has helped detect areas of the model that are wrong and require further improvements to representation of processes. The parameter changes that have been made are backed up with data and so we are putting the right values for the right reason. Tuning can give you the right answer but not always for the right reason, and so should be done carefully.

There is ongoing work to tune certain JULES parameters. Another paper is in review with GMD to evaluate the tuning method (Raoult et al., *in review*). The next step in the model's development will be to combine the tuning with the new trait-based representation presented in this study.

We argue that the extra complication that results from the new PFTs is worth the benefit of having more diverse plant types, which should enable more diverse and specific responses to climate change. A follow-up paper is being finalized which analyzes the impacts of the new PFTs when JULES is run with dynamic vegetation, and results are also improved in this mode.

At the same time, it would be good to evaluate the improvements with extra PFTs compared to just improving parameters with 5 PFTs. As the reviewer suggested, we added a third global experiment to test the 5 PFTs with improved parameters, as in Table SM2. The supplemental material now includes this table plus recommendations for running JULES with 5 PFTs and improved parameters.

**Table SM2**. New trait-based parameters for 5 PFTs that are consistent with TRY data. $N_m$, LMA, and $\gamma_0$ (=1/[leaf lifespan in years]) were calculated directly from the data collected. The slopes and intercept parameters for $V_{cmax}$ ($s_v$ and $i_v$, respectively) were calculated based on the average of observed values available from Kattge et al. (2009).

| | BT | NT | C3 | C4 | SH |
|---|---|---|---|---|---|
| $N_m$ | 0.0185 | 0.0117 | 0.0240 | 0.0113 | 0.0175 |
| LMA | 0.1012 | 0.2240 | 0.0495 | 0.1370 | 0.1023 |
| $s_v$ | 25.48 | 18.15 | 40.96 | 20.48 | 23.15 |
| $i_v$ | 6.12 | 6.32 | 6.42 | 0.00 | 14.71 |
| $V_{cmax,25}$ | 53.84 | 53.88 | 55.08 | 31.71 | 56.15 |
| $T_{off}$ | 5 | -40 | 5 | 5 | -40 |
| $d_T$ | 9 | 9 | 0 | 0 | 9 |
| $\gamma_0$ | 0.25 | 0.25 | 3.0 | 3.0 | 0.66 |
| $\gamma_p$ | 20 | 15 | 20 | 20 | 15 |
| $L_{min}$ | 1 | 1 | 1 | 1 | 1 |
| $L_{max}$ | 9 | 7 | 3 | 3 | 4 |

We also change the following parameters from their default value in Table 1 to make the parameters consistent with JULES9$_{ALL}$:

| | BT | NT | C$_3$ | C$_4$ | SH |
|---|---|---|---|---|---|
| $D_{crit}$ | 0.09 | 0.06 | 0.051 | 0.075 | 0.037 |
| $f_0$ | 0.875 | 0.875 | 0.931 | 0.800 | 0.950 |
| $f_d$ | 0.010 | 0.015 | 0.019 | 0.019 | 0.015 |
| $rootd$ | 3 | 2 | 0.5 | 0.5 | 1 |
| $T_{low}$ | 5 | 0 | 10 | 13 | 0 |
| $T_{opt}$ | 39 | 32 | 28 | 41 | 32 |
| $T_{upp}$ | 43 | 36 | 32 | 45 | 36 |
| $\alpha$ | 0.08 | 0.08 | 0.06 | 0.04 | 0.08 |
| $\mu_{rl}$ | 0.67 | 0.67 | 0.72 | 0.72 | 0.67 |
| $\mu_{sl}$ | 0.10 | 0.10 | 1.00 | 1.00 | 0.10 |

The new experiment is called JULES5$_{ALL}$, since it included as many parameter updates as possible to give a fair comparison between JULES with 5 PFTs and JULES9$_{ALL}$. Most of the differences in GPP and NPP between JULES5$_{ALL}$ and JULES9$_{ALL}$ were in the tropics. The global GPP was high (135 Pg C yr$^{-1}$) in JULES5$_{ALL}$, primarily because $V_{cmax}$ for the average broadleaf tree (53.84 μmol m$^{-2}$ s$^{-1}$) was much higher than for the tropical broadleaf evergreen PFT (41.17 μmol m$^{-2}$ s$^{-1}$). Although tropical GPP was higher in JULES5$_{ALL}$ compared to JULES9$_{ALL}$, the NPP was lower and closer to the values from MODIS NPP. The reason was the differences in leaf nitrogen, which increased respiratory costs in JULES5$_{ALL}$ compared to JULES9$_{ALL}$. Both $N_A$ and $N_m$ were higher for the broadleaf tree PFT (1.87 g N m$^{-2}$ and 0.0185 g N g$^{-1}$, respectively) than for the tropical evergreen broadleaf tree PFT (1.77 g N m$^{-2}$ and 0.0170 g N g$^{-1}$, respectively).

We have added an explanation of this simulation, its results, and implications in the manuscript at Lines: 439-442, 679-687, and 759-767. Also the global results are shown in a new figure (Fig. 9) and summarized on a per-biome basis in Table 6.

**Table 6a**. Area-weighted GPP from each biome (g C m$^{-2}$ yr$^{-1}$). The biome total GPP from MTE is given in Pg C yr$^{-1}$ to give perspective of each biome's role in the global total.

| Biome | JULES5 | JULES9 | JULES5-ALL | MTE | MTE total |
|---|---|---|---|---|---|
| Tropical forest | 2403±217 | 2295±191 | 2505±217 | 2244±297 | 49.9 |
| Tropical forest: Only BET-Tr. | 2924±144 | 2955±147 | 3279±178 | 2790±273 | |
| Tropical savannah | 1355±244 | 1268±223 | 1320±237 | 1111±257 | 21.9 |
| Extratropical mixed forests | 947±147 | 1082±158 | 1119±167 | 1119±212 | 2.9 (13.4*) |
| Boreal and coniferous forests | 514±99 | 597±118 | 645±122 | 650±203 | 12.1 |
| Temperate grasslands | 420±145 | 465±138 | 477±140 | 509±184 | 8.1 |
| Deserts and shrublands | 82±48 | 91±46 | 91±47 | 283±200 | 4.9 |
| Tundra | 86±20 | 94±20 | 101±20 | 279±233 | 1.9 |
| Mediterranean Woodlands | 324±147 | 407±136 | 405±140 | 510±190 | 1.5 |

*Value for EMF biome when agricultural mask is not applied.

**Table 6b**. Area-weighted NPP from each biome (g C m$^{-2}$ yr$^{-1}$).

| Biome | JULES5 | JULES9 | JULES5-ALL | MODIS17 |
|---|---|---|---|---|
| Tropical forest | 956±144 | 1007±125 | 951±143 | 786±352 |
| Only BET-Tr. | 1141±101 | 1233±103 | 1109±126 | 929±315 |
| Tropical savannah | 527±158 | 591±143 | 584±152 | 451±319 |
| Extratropical mixed forests | 586±93 | 631±104 | 640±110 | 563±231 |
| Boreal and coniferous forests | 307±65 | 358±77 | 385±80 | 350±155 |
| Temperate grasslands | 180±94 | 243±89 | 242±90 | 304±247 |
| Deserts and shrublands | 16±29 | 35±29 | 33±29 | 111±133 |
| Tundra | 52±14 | 61±13 | 65±13 | 136±94 |
| Mediterranean Woodlands | 118±94 | 201±89 | 195±89 | 324±184 |

**Further referee comments:**

Experiments 4+ are discussed before experiments 1 to 3 in the text. It would be easier to follow if all the experiments were described in the same way and in the same order. Perhaps move the method around line 515 from the results section to before the first mention of experiment 4?

We switched sections 2.3.1 and 2.3.2 so the Experiments are described in Section 2 in the correct order. We also added further explanation of these experiments at the beginning of Section 2 (~Line 170-172), and of the calculation of the relative statistics (Line 469-470). However now Table 4 is mentioned before Table 3 so these are switched throughout the manuscript.

Table SM 2 gives tuned parameters for the tuned 5 PFT JULES, but I cannot find a reference to that in the text. Is there a missing section?

Yes Table SM2 should be referenced in the text. Thank you for catching this. It is now referred to at Line 176-177. Also extra discussion is added to the supplementary material (see page 3 of SM).

"and updated the model phenology to include a trade-off between leaf lifespan and leaf mass per unit area." - Does your improvement not just change the leaf turnover rate and its impact on the carbon flux rather than the phenology, which is still controlled in the same way as traditional JULES?

It is true that the equations controlling phenology in JULES (Eq. 15-16) were not changed. However, changing the temperature threshold, $T_{off}$, did change the timing of when leaves grow in the fall and senesce in the fall. The trade-off referred to here is included in JULES by increasing leaf growth in the spring ($\gamma_p$) and turnover rates in the fall ($\gamma_0$) for leaves with low LMA, while maintaining low turnover rates for the thicker, longer-lived leaves. However it could be misleading to say the phenology was updated since no structural changes were made to the model so we have reworded this sentence in the abstract.

General comment for reviewers:

Note that in two places we have changed "tuning" to "calibration" as the parameter changes were not really tuned in a strict sense (Line 167, Line 1204). There is a tool for tuning parameters in JULES (adJULES, Raoult et al., 2016), but this was not used in this study. So we believe the change from "tuning" to "calibration" is a more appropriate description of what was done, and will avoid confusion between what can be done with adJULES and the techniques used in this study (adjustment of parameters to correct biases, or more frequently new parameters based on data and literature review). The justification for each parameter change has already been provided in the Methods section.

Lines 167-168: Updated parameters were based on review of literature

---

## Author Comment (AC2) · 26 Apr 2016

Please find attached the revised manuscript with changes highlighted in red.

Please also note the supplement to this comment:
http://www.geosci-model-dev-discuss.net/gmd-2016-22/gmd-2016-22-AC2-supplement.zip
* * *

---

## Author Response (AR1)

Responses to Anonymous Referee #1 on submission to Geoscientific Model Development Discussion (doi:10.5194/gmd-2016-22).

Submitted by Anna Harper on behalf of myself and my co-authors.

We thank the referee for your helpful comments on the manuscript and for taking the time to review it. Below we include the referee comments in black and our responses in red. The supplement contains a revised manuscript with red indicating changed sections. All line numbers refer to that version of the manuscript. Note the revised manuscript also includes some edits of minor errors (all in red for traceability).

The paper "Improved representation of plant functional types and physiology in the Joint UK Land Environment Simulator (JULES v4.2) using plant trait information" present several improvements of the JULES DGVM. these improvement are first based on increasing the number of PFTs from 5 to 9 to better represent the different types of leaves in the leaf economic spectrum including deciduous and evergreen trees and a separation between climate zones. Second improvement was done in estimation of leaf photosynthesis from leaf nitrogen and improvement of phenology considering a more realistic leaf longevity.

This is an important paper that allow to follow recent developments of the JULES model and perfectly fit to the objective GMD. The changes are sufficiently important to justify the publication of a paper. The paper is well written with a convincing evaluation of new model performances both at site level and at global scale. The results show a clear improvement of the model at different scales. For all these reasons I recommend the paper for publication. Here after are just some minor comments that could help to improve the manuscript:

- There is no real justification of the choice of 9 PFTs except as a minimum to represent the main leaves forms. Obviously, for technical reasons, the number of PFTs cannot be increase indefinitely and then a compromise should be find but it would be interesting to see if including a higher number of PFT should also give higher performances?
One way could be to look to the differences between simulated GPP and NPP and respectively Jung and MODIS maps for each pixels and each PFT. Then we could see if there is spatially coherent systematic bias that could show possible new PFT separation.

It's true that the choice of PFTs is subjective. The 9 PFTs were chosen as they represent the range of deciduous and evergreen plant types with minimal externally determined bioclimatic limits. The distinction between tropical and temperate broadleaf evergreen trees exists to account for the important differences between these types of trees, as described in the Introduction. In particular, measured $V_{cmax}$ for a given leaf N per unit area ($N_A$) can be lower in tropical evergreen trees than in temperate broadleaf evergreen trees (Kattge et al., 2009), resulting in lower $V_{cmax}$ and maximum assimilation rates for tropical forests. We have added these justifications in the Discussion (Lines 759-763).

Previously JULES was hard-wired for 5 PFTs. An important step in going from 5 to 9 PFTs was removing this hardwiring. Now users can define the number of PFTs, so the 9 documented in this paper are a recommendation but can be adjusted in the future. This is now mentioned at Line 754-756.

However, it is a good suggestion to evaluate in a more objective way if an appropriate number of PFTs has been chosen. One logical way to further subdivide the PFTs is based on the biome maps, which is similar to the reviewer's suggestion. The analysis in the manuscript was based on an

original data set of 14 biomes, where some biomes were combined for a total of 7 biomes. In a new figure (Fig. SM6) we show the biases in JULES5 and JULES9 for 11 of the original 14 biomes (3 biomes are very small and had no visible differences in the maps: Tropical Coniferous forests, Mangroves, and Flooded grasslands/savannas). The area-weighted RMSE is given in the top left of each map. Some biomes do not show an improvement in JULES9 and this gives some indication where extra PFTs might improve the simulation: for example the Boreal Forests/Tiaga; Tundra; Mediterranean woodlands; and Desert/Xeric Shrublands. Also the biases are still very high for the tropical/subtropical forest and grassland biomes. These regions broadly agree with what was mentioned in Section 5.2. We have added a more specific recommendation for development in these regions at Line 770-772.

However we also caution against defining too many PFTs, as there is already overlap between the $N_m$ and LMA traits (Fig. 1c). In developing new PFTs it would be ideal to determine definitions that result in distinctive sets of traits. Future work will address the possibility of more PFTs or improved processes with the new framework for flexible PFTs in JULES.

minor comments on figures:
- Figure 7: what represent the grey zone ?
This is the standard deviation of the observed fluxes, based on the monthly means from all months. This information has been added to the caption.
- Figure 8: The figure is difficult to read mainly because this is tiny figures. Should it be possible to split it to have larger figures ?
We rearranged this figure so the individual panels can be larger.

General comment for reviewers:

Note that in two places we have changed "tuning" to "calibration" as the parameter changes were not really tuned in a strict sense (Line 167, Line 1204). There is a tool for tuning parameters in JULES (adJULES, Raoult et al., 2016), but this was not used in this study. So we believe the change from "tuning" to "calibration" is a more appropriate description of what was done, and will avoid confusion between what can be done with adJULES and the techniques used in this study (adjustment of parameters to correct biases, or more frequently new parameters based on data and literature review). The justification for each parameter change has already been provided in the Methods section.

Lines 167-168: Updated parameters were based on review of literature

[Figure]

**Figure 7.** Area-averaged seasonal cylces of GPP from the biomes shown in Fig. 3, comparing JULES5, JULES9, and the Jung et al. (2011) MTE. Also shown are the temperature and precipitation from the CRU-NCEP dataset used to force the JULES simulations. The gray shading in the GPP plots shows the MTE GPP ±1 standard deviation based on the area-averaged standard deviations of monthly fluxes for each grid cell.

—— MTE

—— JULES9-ALL

—— JULES5

[Figure]

**Figure 8.** Global maps of carbon cycle fluxes from 2000-2012. The observation sources are: MTE (GPP), and MODIS MOD17 (NPP, 2000-2013).

[Figure]

**Figure S6a.** Differences between JULES5 and JULES9 and the MTE GPP for each of the 11 major biomes from the WWF database (biomes with area > 1,000 km²). The area-average root mean square error is given for each map.

[Figure]

**Figure S6b.** Differences between JULES5 and JULES9 and the MTE GPP for each of the 11 major biomes from the WWF database (biomes with area > 1,000 km²). The area-average root mean square error is given for each map.

[Figure]

**Figure S6c.** Differences between JULES5 and JULES9 and the MTE GPP for each of the 11 major biomes from the WWF database (biomes with area > 1,000 km$^2$). The area-average root mean square error is given for each map.

Responses to Anonymous Referee #2 on submission to Geoscientific Model Development Discussion (doi:10.5194/gmd-2016-22).

Submitted by Anna Harper on behalf of myself and my co-authors.

We thank the referee for your helpful comments on the manuscript and for taking the time to review it. Below we include the referee comments in black (with the specific questions addressed in bold), and our responses in red. The supplement contains a revised manuscript with red indicating changed sections. All line numbers refer to that version of the manuscript. Note the revised manuscript also includes some edits of minor errors (all in red for traceability).

The authors present a version of the JULES land surface model with a more detailed dynamic vegetation model and show that this gives more accurate carbon fluxes than the traditional version of JULES. It is of great interest and should be published. **My only question is whether you could you have got the same answer by tuning the old version of JULES?** Adding extra PFTs will cause greater complication than tuning parameters, especially when competition between PFTs is turned on. You say you corrected known biases in the model. **Did these same biases get corrected in the original, 5 PFT version, or just the new version?** If not, I think you should have added an extra experiment to assess the relative impact on the flux from adding the additional PFTs and the tuning. **Would just correcting the 5 PFT JULES have had the same impact as adding extra PFTs?** I think that some discussion of this, and ideally an extra experiment, is needed.

First, we address the question of tuning. In this study, we have used observations to constrain the model. This has improved the model and it has helped detect areas of the model that are wrong and require further improvements to representation of processes. The parameter changes that have been made are backed up with data and so we are putting the right values for the right reason. Tuning can give you the right answer but not always for the right reason, and so should be done carefully.

There is ongoing work to tune certain JULES parameters. Another paper is in review with GMD to evaluate the tuning method (Raoult et al., *in review*). The next step in the model's development will be to combine the tuning with the new trait-based representation presented in this study.

We argue that the extra complication that results from the new PFTs is worth the benefit of having more diverse plant types, which should enable more diverse and specific responses to climate change. A follow-up paper is being finalized which analyzes the impacts of the new PFTs when JULES is run with dynamic vegetation, and results are also improved in this mode.

At the same time, it would be good to evaluate the improvements with extra PFTs compared to just improving parameters with 5 PFTs. As the reviewer suggested, we added a third global experiment to test the 5 PFTs with improved parameters, as in Table SM2. The supplemental material now includes this table plus recommendations for running JULES with 5 PFTs and improved parameters.

**Table SM2**. New trait-based parameters for 5 PFTs that are consistent with TRY data. $N_m$, LMA, and $\gamma_0$ (=1/[leaf lifespan in years]) were calculated directly from the data collected. The slopes and intercept parameters for $V_{cmax}$ ($s_v$ and $i_v$, respectively) were calculated based on the average of observed values available from Kattge et al. (2009).

|  | BT | NT | C3 | C4 | SH |
|---|---|---|---|---|---|
| $N_m$ | 0.0185 | 0.0117 | 0.0240 | 0.0113 | 0.0175 |
| LMA | 0.1012 | 0.2240 | 0.0495 | 0.1370 | 0.1023 |
| $s_v$ | 25.48 | 18.15 | 40.96 | 20.48 | 23.15 |
| $i_v$ | 6.12 | 6.32 | 6.42 | 0.00 | 14.71 |
| $V_{cmax,25}$ | 53.84 | 53.88 | 55.08 | 31.71 | 56.15 |
| $T_{off}$ | 5 | -40 | 5 | 5 | -40 |
| $d_T$ | 9 | 9 | 0 | 0 | 9 |
| $\gamma_0$ | 0.25 | 0.25 | 3.0 | 3.0 | 0.66 |
| $\gamma_p$ | 20 | 15 | 20 | 20 | 15 |
| $L_{min}$ | 1 | 1 | 1 | 1 | 1 |
| $L_{max}$ | 9 | 7 | 3 | 3 | 4 |

We also change the following parameters from their default value in Table 1 to make the parameters consistent with JULES9$_{ALL}$:

|  | BT | NT | C3 | C4 | SH |
|---|---|---|---|---|---|
| $D_{crit}$ | 0.09 | 0.06 | 0.051 | 0.075 | 0.037 |
| $f_0$ | 0.875 | 0.875 | 0.931 | 0.800 | 0.950 |
| $f_d$ | 0.010 | 0.015 | 0.019 | 0.019 | 0.015 |
| $rootd$ | 3 | 2 | 0.5 | 0.5 | 1 |
| $T_{low}$ | 5 | 0 | 10 | 13 | 0 |
| $T_{opt}$ | 39 | 32 | 28 | 41 | 32 |
| $T_{upp}$ | 43 | 36 | 32 | 45 | 36 |
| $\alpha$ | 0.08 | 0.08 | 0.06 | 0.04 | 0.08 |
| $\mu_{rl}$ | 0.67 | 0.67 | 0.72 | 0.72 | 0.67 |
| $\mu_{sl}$ | 0.10 | 0.10 | 1.00 | 1.00 | 0.10 |

The new experiment is called JULES5$_{ALL}$, since it included as many parameter updates as possible to give a fair comparison between JULES with 5 PFTs and JULES9$_{ALL}$. Most of the differences in GPP and NPP between JULES5$_{ALL}$ and JULES9$_{ALL}$ were in the tropics. The global GPP was high (135 Pg C yr$^{-1}$) in JULES5$_{ALL}$, primarily because $V_{cmax}$ for the average broadleaf tree (53.84 μmol m$^{-2}$ s$^{-1}$) was much higher than for the tropical broadleaf evergreen PFT (41.17 μmol m$^{-2}$ s$^{-1}$). Although tropical GPP was higher in JULES5$_{ALL}$ compared to JULES9$_{ALL}$, the NPP was lower and closer to the values from MODIS NPP. The reason was the differences in leaf nitrogen, which increased respiratory costs in JULES5$_{ALL}$ compared to JULES9$_{ALL}$. Both $N_A$ and $N_m$ were higher for the broadleaf tree PFT (1.87 g N m$^{-2}$ and 0.0185 g N g$^{-1}$, respectively) than for the tropical evergreen broadleaf tree PFT (1.77 g N m$^{-2}$ and 0.0170 g N g$^{-1}$, respectively).

We have added an explanation of this simulation, its results, and implications in the manuscript at Lines: 439-442, 679-687, and 759-767. Also the global results are shown in a new figure (Fig. 9) and summarized on a per-biome basis in Table 6.

**Table 6a**. Area-weighted GPP from each biome (g C m$^{-2}$ yr$^{-1}$). The biome total GPP from MTE is given in Pg C yr$^{-1}$ to give perspective of each biome's role in the global total.

| Biome | JULES5 | JULES9 | JULES5-ALL | MTE | MTE total |
|---|---|---|---|---|---|
| Tropical forest | 2403±217 | 2295±191 | 2505±217 | 2244±297 | 49.9 |
| Tropical forest: Only BET-Tr. | 2924±144 | 2955±147 | 3279±178 | 2790±273 | |
| Tropical savannah | 1355±244 | 1268±223 | 1320±237 | 1111±257 | 21.9 |
| Extratropical mixed forests | 947±147 | 1082±158 | 1119±167 | 1119±212 | 2.9 (13.4*) |
| Boreal and coniferous forests | 514±99 | 597±118 | 645±122 | 650±203 | 12.1 |
| Temperate grasslands | 420±145 | 465±138 | 477±140 | 509±184 | 8.1 |
| Deserts and shrublands | 82±48 | 91±46 | 91±47 | 283±200 | 4.9 |
| Tundra | 86±20 | 94±20 | 101±20 | 279±233 | 1.9 |
| Mediterranean Woodlands | 324±147 | 407±136 | 405±140 | 510±190 | 1.5 |

*Value for EMF biome when agricultural mask is not applied.

**Table 6b**. Area-weighted NPP from each biome (g C m$^{-2}$ yr$^{-1}$).

| Biome | JULES5 | JULES9 | JULES5-ALL | MODIS17 |
|---|---|---|---|---|
| Tropical forest | 956±144 | 1007±125 | 951±143 | 786±352 |
| Only BET-Tr. | 1141±101 | 1233±103 | 1109±126 | 929±315 |
| Tropical savannah | 527±158 | 591±143 | 584±152 | 451±319 |
| Extratropical mixed forests | 586±93 | 631±104 | 640±110 | 563±231 |
| Boreal and coniferous forests | 307±65 | 358±77 | 385±80 | 350±155 |
| Temperate grasslands | 180±94 | 243±89 | 242±90 | 304±247 |
| Deserts and shrublands | 16±29 | 35±29 | 33±29 | 111±133 |
| Tundra | 52±14 | 61±13 | 65±13 | 136±94 |
| Mediterranean Woodlands | 118±94 | 201±89 | 195±89 | 324±184 |

**Further referee comments:**

Experiments 4+ are discussed before experiments 1 to 3 in the text. It would be easier to follow if all the experiments were described in the same way and in the same order. Perhaps move the method around line 515 from the results section to before the first mention of experiment 4?

We switched sections 2.3.1 and 2.3.2 so the Experiments are described in Section 2 in the correct order. We also added further explanation of these experiments at the beginning of Section 2 (~Line 170-172), and of the calculation of the relative statistics (Line 469-470). However now Table 4 is mentioned before Table 3 so these are switched throughout the manuscript.

Table SM 2 gives tuned parameters for the tuned 5 PFT JULES, but I cannot find a reference to that in the text. Is there a missing section?

Yes Table SM2 should be referenced in the text. Thank you for catching this. It is now referred to at Line 176-177. Also extra discussion is added to the supplementary material (see page 3 of SM).

"and updated the model phenology to include a trade-off between leaf lifespan and leaf mass per unit area." - Does your improvement not just change the leaf turnover rate and its impact on the carbon flux rather than the phenology, which is still controlled in the same way as traditional JULES?

It is true that the equations controlling phenology in JULES (Eq. 15-16) were not changed. However, changing the temperature threshold, $T_{off}$, did change the timing of when leaves grow in the fall and senesce in the fall. The trade-off referred to here is included in JULES by increasing leaf growth in the spring ($\gamma_p$) and turnover rates in the fall ($\gamma_0$) for leaves with low LMA, while maintaining low turnover rates for the thicker, longer-lived leaves. However it could be misleading to say the phenology was updated since no structural changes were made to the model so we have reworded this sentence in the abstract.

General comment for reviewers:

Note that in two places we have changed "tuning" to "calibration" as the parameter changes were not really tuned in a strict sense (Line 167, Line 1204). There is a tool for tuning parameters in JULES (adJULES, Raoult et al., 2016), but this was not used in this study. So we believe the change from "tuning" to "calibration" is a more appropriate description of what was done, and will avoid confusion between what can be done with adJULES and the techniques used in this study (adjustment of parameters to correct biases, or more frequently new parameters based on data and literature review). The justification for each parameter change has already been provided in the Methods section.

Lines 167-168: Updated parameters were based on review of literature

[revised manuscript text omitted]

This supplement contains:
S1. Additional photosynthesis equations for the JULES model
S2. Data for $N_{mass}$, LMA, and LL.
S3. Energy and respiration results at the Fluxnet sites
Supplemental Tables SM1-SM2
Supplemental Figures S1-S6
Supplemental Data in the zip file TRY_data_PFTs.zip contains summary of data for the standard 5 PFTs and new 9 PFTs.

**S1. Additional photosynthesis equations**

In JULES, leaf-level photosynthesis (Collatz et al 1991:1992) is calculated based on the limiting factor of three potential photosynthesis rates.

1. A light-limited rate, $W_l$:

$$W_l = \alpha(1-\omega)I_{par}\left(\frac{c_i-\Gamma}{c_i+2\Gamma}\right) \quad for\ C3\ plants \tag{A.1}$$

$$W_l = \alpha(1-\omega)I_{par} \qquad for\ C4\ plants \tag{A.2}$$

where $\alpha$ is the quantum efficiency of photosynthesis (mol $CO_2$ mol $PAR^{-1}$) and $\omega$ is the leaf scattering coefficient for PAR. $I_{par}$ is the photosynthetically active radiation hitting the leaf (mol $m^{-2}$ $s^{-1}$), $\Gamma$ is the $CO_2$ compensation point in the absence of mitochondrial respiration (Pa), and $c_i$ is the internal $CO_2$ concentration (Pa).

2. A Rubisco-limited rate, $W_c$:

$$W_c = V_{cmax}\left(\frac{c_i-\Gamma}{c_i+K_c\left(1+{O_a}/{K_o}\right)}\right) for\ C3\ plants \tag{A.3}$$

$$W_c = V_{cmax} \qquad for\ C4\ plants \tag{A.4}$$

where $K_O$ and $K_C$ are the Michaelis-Menten parameters for $O_2$ and $CO_2$, respectively, and $V_{cmax}$ is the maximum rate of carboxylation of Rubisco ($\mu$mol $CO_2$ $m^{-2}$ $s^{-1}$).

3. A rate of transport of photosynthetic products for C3 plants, and PEPCarboxylase limitation for C4 plants, $W_e$:

$$W_e = 0.5V_{cmax} \qquad\qquad for\ C3\ plants \qquad\qquad\qquad (A.5)$$

$$W_e = 2x10^4\ V_{cmax}\ \frac{c_i}{P_*} \qquad for\ C4\ plants \qquad\qquad\qquad (A.6)$$

where $P_*$ is the surface air pressure (Pa).

**S2. Data for N$_{mass}$, LMA, and LL.** The table shows the data sources for the TRY data used in

this study. For each source, the number of measurements for each source is provided for

N$_{mass}$/specific leaf area (SLA) pairs and for leaf lifespan.

| Ref. | Contact | N$_{mass}$ + SLA | Leaf lifespan |
|---|---|---|---|
| Atkin et al., 1997; Campbell et al., 2007 | Owen Atkin | 218 | |
| Xu and Baldocchi, 2003 | Dennis Baldocchi | 468 | |
| Cavender-Bares et al., 2006 | Jeannine Cavender-Bares | x | |
| unpublished | F. Stuart III Chapin | 50 | 48 |
| Cornelissen et al., 2004; Cornelissen et al., 2003, 1996; Diaz et al., 2004; Quested et al., 2003 | Johannes Cornelissen | 690 | 161 |
| | Will Cornwell (+David Ackerly) | 53 | |
| Díaz et al., 2004 (maybe didn't use); Diaz et al. 2010 (definitely used) | Sandra Díaz | 70 | |
| Han et al., 2005; He et al., 2006, 2007 | Jingyun Fang | 148 | |
| Freschet and Cornelissen, 2010; Freschet et al., 2010 | Gregoire Freschet (+Hans Cornelissen) | 40 | |
| | Eric Garnier (+ Sandra Lavorel) | 966 | |
| Kattge et al., 2009 | Jens Kattge* | 1326 | 204 |
| Kurokawa and Nakashizuka, 2008 | Hiroko Kurokawa | 399 | 89 |
| | Daniel Laughlin | 139 | |
| Niinemets, 1999; Niinemets, 2001 | Ülo Niinemets | 264 | 33 |
| Ordoñez et al., 2010; Ordoñez et al., 2010 | Jenny Ordoñez (+Peter van Bodegom) | 282 | |
| Ogaya and Peñuelas, 2007a, 2007b, 2003; | Josep Peñuelas | 808 | |

| | | | |
|---|---|---|---|
| Ogaya, 2006; Sardans et al., 2008 | | | |
| Poorter et al., 2009; 2006 | Lourens Poorter | | x |
| Reich et al., 2008, 2009 | Peter Reich | 720 | 199 |
| Cornwell et al., 2007 | Lawren Sack | 30 | |
| Shipley and Lechowicz 2000, Ecoscience, 7:183-194
2. Meziane and Shipley, 1999, Plant Cell and Environment | Bill Shipley | 603 | |
| | Enio Sosinski | 66 | |
| Soudzilovskaia et al, 2013, PNAS | Nadia Soudzilovskaia | 155 | |
| | Peter van Bodegom | x | x |
| Wright et al., 2011 | S. Joseph (Joe) Wright | 204 | |
| Wright et al., 2006, 2004 | Ian Wright | 1673 | 442 |

Data was analysed to calculate the parameters $N_m$ and LMA for the new 9 PFTs (Table 2) and for the original 5 PFTs (Table S2). For JULES users who wish to run the model with the standard 5 PFTs but with updated parameters, the recommended parameters are given in Table S2. Additionally, it is important to set both 'l_trait_phys' and 'l_ht_compete' to True in the jules_vegetation namelist. The former will switch to the trait-based physiology discussed in the main text, and the latter will allow for flexible number of PFTs to compete if the dynamic vegetation mode is used. Results of JULES with the new 9 PFTs and the height-based competition will be discussed in a follow up paper.

**S3. Energy and respiration results at the Fluxnet sites**

The TRY-based parameters give a lower $N_{root}$ and $N_{stem}$ for all PFTs, and a lower $N_{leaf}$ for all PFTs except for $C_3$ grass and BET-Te (Fig. S2). For $C_3$ grass, the simulated $N_{leaf}$ in JULES9 is higher during the winter than in JULES5, since a moderate LAI is maintained due to the new phenology.

$N_{leaf}$ is higher for BET-Te due to thicker leaves than previously. The respiration fluxes at BR-Ma2 provide a good example of the impacts of the new PFT parameters (Fig. SM3). The lower $N_{stem}$ and $N_{root}$ in JULES9 compared to JULES5 (Fig. SM2) reduced simulated $R_{pm}$ (Eqn. 8). As a result, the average increase in plant respiration (46 g C m$^{-2}$ yr$^{-1}$) was much smaller than the increase in GPP (377 g C m$^{-2}$ yr$^{-1}$), and NPP increased from 867 g C m$^{-2}$ yr$^{-1}$ in JULES5 to 1198 g C m$^{-2}$ yr$^{-1}$ in JULES9, which was higher than the observed value at Manaus of 1011±140 g C m$^{-2}$ yr$^{-1}$ (Mahli et al., 2009). At Santarem, NPP also increased, but it was lower than the observed value of 1440 g C m$^{-2}$ yr$^{-1}$.

The higher ecosystem respiration in JULES9 compared to JULES5 that accompanied the increased GPP was less realistic (in terms of RMSE), with 3 exceptions (Fig. SM3), but the seasonal cycle of total respiration was improved at 8 sites. The RMSE decreased at ES-ES1, where lower GPP and R in the fall and winter were more realistic, at JP-Tom, where the switch from generic needle leaf to deciduous needle leaf improved all aspects of the simulation, and at FI-Kaa, where the new phenology of grass also improved the simulation.

In JULES, latent heat flux (LE) is due to evaporation from water stored on the canopy, evaporation of water from the top layer of soil, transpiration through the stomata, and sublimation of snow. The seasonal cycle of LE was improved at nine sites, however $r$ decreased (by <0.03) at BR-Sa1, ES-ES1, BR-Sa3, US-Bo1, and US-FPe, comparing JULES9$_{ALL}$ to JULES5. The RMSE increased by >4 W m$^{-2}$ in JULES9$_{TRY}$ compared to JULES5 at DE-Tha and FI-Hyy, and RMSE increased by a further 4 W m$^{-2}$ at DE-Tha when the photosynthesis/respiration parameters were added due to the higher GPP and stomatal conductance. However, the correlation was >0.91 for both sites. At some forest sites, simulated LE (SH) was too high (low) during the winter and spring (DE-Tha, US-Ha1, and US-MMS), however the LE component contributing to the high bias is site-dependent. For example, from Jan.-Mar. the largest source of LE is evaporation from snow/ice ($E_i$) at Harvard,

canopy evaporation and $E_i$ at Tharandt, and soil evaporation/transpiration at Morgan Monroe. These springtime errors were not affected by the new PFTs. Another consistent bias in the forests was high mid-summer LE (De-Tha, FI-Hyy, US-Ha1, and US-MMS), which in this case always results from the soil evaporation/transpiration. Because the new PFTs tend to increase GPP and stomatal conductance, the errors in summer LE are higher.

**Table SM1.** List of parameters and symbols in the text.

| Symbol | Units | Equation | Description | Default Value[a] |
|---|---|---|---|---|
| $A_l$ | kg C m$^{-2}$ s$^{-1}$ | 5 | Leaf-level photosynthesis | |
| $a_{wl}$ | kg C m$^{-2}$ | 24 | Allometric coefficient | |
| $a_{ws}$ | -- | 24 | Ratio of total to respiring stem carbon | |
| $b_{wl}$ | -- | 24 | Allometric exponent | 1.667 |
| $C_i$ | Pa | 6 | Internal leaf $CO_2$ concentration | |
| $C_{mass}$ | kg C [kg biomass]$^{-1}$ | 23 | Leaf carbon concentration per unit mass | 0.5 for this study |
| $C_s$ | Pa | 6 | Leaf surface $CO_2$ concentration | |
| $D_{crit}$ | kg kg$^{-1}$ | 7 | Critical humidity deficit | |
| $d_T$ | -- | 16 | Rate of change of leaf turnover with temperature | |
| $f_0$ | -- | 7 | Stomatal conductance parameter | |
| $f_d$ | -- | 4 | Leaf dark respiration coefficient | |
| $g_s$ | m s$^{-1}$ | 6 | Leaf-level stomatal conductance | |
| $i_v$ | µmol $CO_2$ m$^{-2}$ s$^{-1}$ | 19 | Intercept for relationship between $N_{area}$ and $V_{cmax,25}$ | |
| $k_n$ | -- | 3, 20 | Extinction coefficient for nitrogen | 0.78 |
| $h$ | m | 13, 23, 24 | Canopy height | |
| $L_{bal}$ | m$^2$ m$^{-2}$ | 12, 13, 22-24 | Balanced leaf area index (maximum LAI given the plant's height) | |
| $L_{max}$ | m$^2$ m$^{-2}$ | | Maximum LAI | |
| $L_{min}$ | m$^2$ m$^{-2}$ | | Minimum LAI | |
| LMA | kg m$^{-2}$ | 18, 21, 22 | Leaf mass per unit area (new parameter) | |
| $N_a$ | kg N m$^{-2}$ | 18 | Leaf nitrogen per unit area | |
| $n_{eff}$ | mol $CO_2$ m$^{-2}$ s$^{-1}$ kg C [kg N]$^{-1}$ | 3 | Constant relating leaf nitrogen to Rubisco carboxylation capacity | |
| $N_{l0}$ | kg N [kg C]$^{-1}$ | 3 | Top leaf nitrogen concentration (old parameter, mass basis) | |
| $N_m$ | kg N kg$^{-1}$ | 18, 21-23 | Top leaf nitrogen concentration (new parameter) | |
| $N_l$ | kg N m$^{-2}$ | 11, 21 | Total leaf nitrogen concentration | |
| $N_r$ | kg N m$^{-2}$ | 12, 22 | Total root nitrogen concentration | |
| $N_s$ | kg N m$^{-2}$ | 13, 23 | Total stem nitrogen concentration | |
| $p$ | -- | 17 | Phenological state (LAI/$L_{bal}$) | |
| $Q_{10,leaf}$ | -- | 2 | Constant for exponential term in temperature function of $V_{cmax}$ | 2 |
| $R_a$ | kg C m$^{-2}$ s$^{-1}$ | 8 | Total plant autotrophic respiration | |
| $R_d$ | kg C m$^{-2}$ s$^{-1}$ | 4, 5 | Leaf dark respiraiton | |
| $r_g$ | -- | 10 | Growth respiration coefficient | 0.25 |
| rootd | m | | e-folding root depth | |
| $s_v$ | µmol $CO_2$ g N$^{-1}$ s$^{-1}$ | 19 | Slope between $N_{area}$ and $V_{cmax,25}$ | |
| $T_{low}$ | °C | 1 | Upper temperature parameter for $V_{cmax}$ | |
| $T_{off}$ | °C | 16 | Threshold temperature for phenology | |
| $T_{opt}$[b] | °C | | Optimal temperature for $V_{cmax}$ | |
| $T_{upp}$ | °C | 1 | Upper temperature parameter for $V_{cmax}$ | |
| $V_{cmax,25}$ | µmol m$^{-2}$ s$^{-1}$ | 1, 9 | The maximum rate of carboxylation of Rubisco at 25°C | |
| $W$ | kg C m$^{-2}$ s$^{-1}$ | 5 | Smoothed minimum of the potential limiting rates of phososynthesis | |
| $\alpha$ | mol $CO_2$ [mol PAR photons]$^{-1}$ | | Quantum efficiency | |
| $\beta$ | -- | 5 | Soil moisture stress factor | |
| $\Gamma^*$ | Pa | 7 | $CO_2$ compensation point | |
| $\gamma_0$ | [360 days]$^{-1}$ | 16 | Minimum leaf turnover rate | |
| $\gamma_{lm}$ | [360 days]$^{-1}$ | 16 | Leaf turnover rate | |

| | | | | | |
|---|---|---|---|---|---|
| $\gamma_p$ | [360 days]$^{-1}$ | 17 | Leaf growth rate | | 20 |
| $\mu_{rl}$ | -- | 12, 22 | Ratio of nitrogen concentration in roots and leaves | | |
| $\mu_{sl}$ | -- | 13, 23 | Ratio of nitrogen concentration in stems and leaves | | |
| $\eta_{sl}$ | kg C m$^{-2}$ LAI$^{-1}$ | 13, 23 | Live stemwood coefficient | | 0.01 |
| $\sigma_L$ | kg C m$^{-2}$ LAI$^{-1}$ | 11, 12 | Specific leaf density (old parameter) | | |

[a]Default values only provided for non-PFT-dependent parameters.

**Table SM2**. New trait-based parameters for 5 PFTs that are consistent with TRY data and updated parameters used in this study.

| | BT | NT | C3 | C4 | SH |
|---|---|---|---|---|---|
| $N_m$ | 0.0185 | 0.0117 | 0.0240 | 0.0113 | 0.0175 |
| LMA | 0.1012 | 0.2240 | 0.0495 | 0.1370 | 0.1023 |
| $s_v$ | 25.48 | 18.15 | 40.96 | 20.48 | 23.15 |
| $i_v$ | 6.12 | 6.32 | 6.42 | 0.00 | 14.71 |
| $V_{cmax,25}$ | 53.84 | 53.88 | 55.08 | 31.71 | 56.15 |
| $T_{off}$ | 5 | -40 | 5 | 5 | -40 |
| $d_T$ | 9 | 9 | 0 | 0 | 9 |
| $\gamma_0$ | 0.25 | 0.25 | 3.0 | 3.0 | 0.66 |
| $\gamma_p$ | 20 | 15 | 20 | 20 | 15 |
| $L_{min}$ | 1 | 1 | 1 | 1 | 1 |
| $L_{max}$ | 9 | 7 | 3 | 3 | 4 |
| $D_{crit}$ | 0.09 | 0.06 | 0.051 | 0.075 | 0.037 |
| $f_0$ | 0.875 | 0.875 | 0.931 | 0.800 | 0.950 |
| $f_d$ | 0.010 | 0.015 | 0.019 | 0.019 | 0.015 |
| $rootd$ | 3 | 2 | 0.5 | 0.5 | 1 |
| $T_{low}$ | 5 | 0 | 10 | 13 | 0 |
| $T_{opt}$ | 39 | 32 | 28 | 41 | 32 |
| $T_{upp}$ | 43 | 36 | 32 | 45 | 36 |
| $\alpha$ | 0.08 | 0.08 | 0.06 | 0.04 | 0.08 |
| $\mu_{rl}$ | 0.67 | 0.67 | 0.72 | 0.72 | 0.67 |

**Table SM3.** Relationship between WWF ecoregions and the eight biomes used in this study.

| WWF ecoregion | Biome for this study |
|---|---|
| Tropical & Subtropical Moist Broadleaf Forests | Tropical forests |
| Tropical & Subtropical Dry Broadleaf Forests | Tropical forests |
| Tropical & Subtropical Coniferous Forests | Tropical forests |
| Temperate Broadleaf & Mixed Forests | Extratropical mixed forests |
| Temperate Conifer Forests | Boreal and coniferous forests |
| Boreal Forests/Taiga | Boreal and coniferous forests |
| Tropical & Subtropical Grasslands, Savannas & Shrublands | Tropical savannas |
| Temperate Grasslands, Savannas & Shrublands | Temperate grasslands |
| Flooded Grasslands & Savannas | Temperate grasslands |
| Montane Grasslands & Shrublands | Temperate grasslands |
| Tundra | Tundra |
| Mediterranean Forests, Woodlands & Scrub | Mediterranean woodlands |
| Deserts & Xeric Shrublands | Desert |
| Mangroves | Tropical forests |

Supplemental Material:
Figures

[Figure]

**Figure S1.** Daily average GPP versus shortwave radiation on days following rainfall when leaf area index is at or near its seasonal maximum.

[Figure]

**Figure S2.** Monthly mean leaf nitrogen content (scaled by LAI) at nine sites representative of each of the new PFTs.

[Figure]

**Figure S3a.** Monthly mean fluxes of GPP, total ecosystem respiration, NEE, autotrophic respiration, and NPP at two tropical forest sites (BET-tr) and two savannah sites (EvSa=Evergreen Savannah, DeSa=Deciduous Savannah). Observations ± standard deviation from Fluxnet are shown with triangles and vertical lines. All simulations in Table 4 in the main text are shown. Also shown are the daily root mean square error (rmse) based on daily fluxes and the correlation coefficient (r) based on monthly mean fluxes for all years of the simulations. All units are in gC m$^{-2}$ d$^{-1}$.

[Figure]

**Figure S3b.** As in SM3a but for three needleleaf evergreen (NET) sites.

[Figure]

**Figure S3c.** As in SM3a but for four grass sites.

[Figure]

**Figure S3d.** As in SM3a but for two broadleaf deciduous (BDT) sites and one needleleaf deciduous (NDT) site.

[Figure]

**Figure S4a.** Monthly mean fluxes of latent heat, sensible heat, and evaporative fraction (=LE/(SH +LE) at two tropical forest sites (BET-tr) and two savannah sites (EvSa=Evergreen Savannah, DeSa=Deciduous Savannah). Observations ± standard deviation from Fluxnet are shown with triangles and vertical lines.

[Figure]

**Figure S4b.** As in Fig. S4a but for the NET sites.

[Figure]

**Figure S4c.** As in Fig. S4a but for the grass sites.

[Figure]

**Figure S4d.** As in Fig. S4a but for the deciduous tree sites.

[Figure]

**Figure S5.** Seasonal cylces of Reco from the biomes shown in Fig. 3, comparing JULES5, JULES9, and the Jung et al. (2011) MTE. Also shown are the temperature and precipitation from the CRU-NCEP dataset used to force the JULES simulations.

—— MTE

—— JULES9-ALL

—— JULES5

[Figure]

**Figure S6a.** Differences between JULES5 and JULES9 and the MTE GPP for each of the 11 major biomes from the WWF database (biomes with area > 1,000 km²). The area-average root mean square error is given for each map.

[Figure]

**Figure S6b.** Differences between JULES5 and JULES9 and the MTE GPP for each of the 11 major biomes from the WWF database (biomes with area > 1,000 km²). The area-average root mean square error is given for each map.

[Figure]

**Figure S6c.** Differences between JULES5 and JULES9 and the MTE GPP for each of the 11 major biomes from the WWF database (biomes with area > 1,000 km$^2$). The area-average root mean square error is given for each map.